# PCNN: Probable-Class Nearest-Neighbor Explanations Improve Fine-Grained Image Classification Accuracy for AIs and Humans

**Giang (Dexter) Nguyen**                                    *nguyengiangbkhn@gmail.com*
*Computer Science and Software Engineering Department*
*Auburn University*

**Valerie Chen**                                             *vchen2@andrew.cmu.edu*
*Machine Learning Department*
*Carnegie Mellon University*

**Mohammad Reza Taesiri**                                    *mtaesiri@gmail.com*
*Electrical & Computer Engineering Department*
*University of Alberta*

**Anh Totti Nguyen**                                         *anh.ng8@gmail.com*
*Computer Science and Software Engineering Department*
*Auburn University*

**Reviewed on OpenReview:** *https://openreview.net/forum?id=OcFjqiJ98b*

## Abstract

Nearest neighbors (NN) are traditionally used to compute final decisions, e.g., in Support Vector Machines or $k$-NN classifiers, and to provide users with explanations for the model's decision. In this paper, we show a novel utility of nearest neighbors: To improve predictions of a frozen, pretrained image classifier $\mathbf{C}$. We leverage an image comparator $\mathbf{S}$ that (1) compares the input image with NN images from the top-$K$ most probable classes given by $\mathbf{C}$; and (2) uses scores from $\mathbf{S}$ to weight the confidence scores of $\mathbf{C}$ to refine predictions. Our method consistently improves fine-grained image classification accuracy on CUB-200, Cars-196, and Dogs-120. Also, a human study finds that showing users our probable-class nearest neighbors (PCNN) reduces over-reliance on AI, thus improving their decision accuracy over prior work which only shows only the most-probable (top-1) class examples.

## 1 Introduction

$k$-nearest neighbors are traditionally considered explainable classifiers by design Papernot & McDaniel (2018). Yet, only recent human studies have found concrete evidence that showing the NNs to humans improves their decision-making accuracy (Nguyen et al., 2021; Liu et al., 2022; Chen et al., 2023b; Chan et al., 2023; Kenny et al., 2022; 2023; Chiaburu et al., 2024; Nguyen et al., 2024), even more effectively than feature attribution in the image domain (Nguyen et al., 2021; Kim et al., 2022). These studies typically presented users with the input image, a model's top-1 prediction, and the NNs from the top-1 class. However, examples from the top-1 class are not always beneficial to users.

One such setting where top-1 neighbors can actually hinder human decision-making accuracy is in fine-grained image classification. When users are asked to accept or reject the model's decision—a *distinction* task, as shown in Fig. 1(a), examples from the top-1 class easily fooled users into incorrectly accepting wrong predictions at an excessively high rate (e.g., at a rate of 81.5% on CUB-200; Table A6 in Taesiri et al. (2022)).

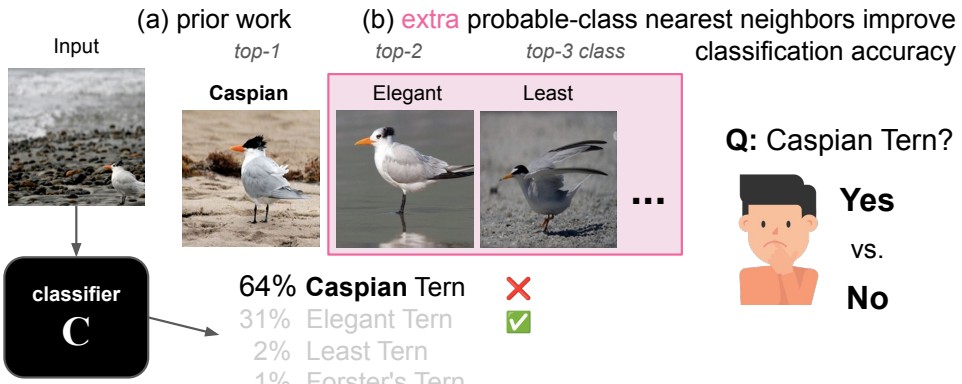

Figure 1: Given an input image $x$ and a black-box, pretrained classifier **C** that predicts the label for $x$. Prior work (a) often shows only the nearest neighbors from the top-1 predicted class as explanations for the decision, which often *fools* humans into accepting *wrong* decisions (here, Caspian Tern) due to the similarity between the input and top-1 class examples. Instead, including extra nearest neighbors (b) from top-2 to top-$K$ classes improves not only human accuracy on this binary distinction task but also AI's accuracy on standard fine-grained image classification tasks (see Fig. 2).

This is because NNs from the top-1 class often looked deceivingly similar to the input (e.g., input vs. Caspian; Fig. 1).

Instead of focusing on top-1 examples, we propose to sample the NN examples from the top-$K$ predicted classes to more fully represent the query and better inform users to make decisions. That is, we propose **Probable-Class Nearest Neighbors** (PCNN), a novel explanation type consisting of $K$ nearest images, where each image is taken from a class among the top-$K$ classes, as illustrated in Fig. 1(b). We show that PCNN not only improves human decisions on the distinction task over showing top-1 class neighbors but can also be leveraged to improve AI-alone accuracy by re-ranking the predicted labels of a pretrained, frozen classifier (Fig. 2).

**Assumptions**   As is the case for many real-world applications, we assume that there exists a pretrained, black-box classifier **C**, e.g., a foundation model (Bommasani et al., 2021), responsible for a large amount of information processing in the pipeline. Due to computation and algorithm constraints, **C** may not be easily re-trained to achieve better accuracy. Therefore, like Bansal et al. (2021), we assume that **C** is frozen—humans or other models would interact with **C** to make final decisions (Fig. 1).

To leverage PCNN for re-ranking **C**'s predicted labels, we train an image comparator **S**, which is a binary classifier that compares the input image with each PCNN example and outputs a sigmoid value that is used to weight the original confidence scores of **C** (Fig. 2). Then, **C** and **S** together form a **C** × **S** model—like a Product of Experts Hinton (1999)—that outperforms **C** alone.

Our experiments on 10 different **C** classifiers across 3 fine-grained classification tasks for bird, car, and dog species reveal:[1].

- Our PCNN-based model consistently improves upon the original **C** accuracy on all three domains: CUB-200, Cars-196, and Dogs-120 (Sec. 4.1).
- Given the same ResNet backbones, our model outperforms similar explainable, prototype-based classifiers including $k$-NN, part-based, and correspondence-based classifiers on all three datasets (Sec. 4.5).

---

[1]Code and data are available at `https://github.com/anguyen8/nearest-neighbor-XAI`

- Interestingly, even without further training the comparator **S** on a new pretrained **C**—we still obtain large gains (up to +23.38 points) when combining a well-trained comparator **S** with an arbitrary **C** model (Sec. 4.4).
- A 60-user study finds that PCNN explanations, compared with top-1 class examples, reduce over-reliance on AI, thus improving user performance on the distinction task by almost 10 points (54.55% vs. 64.58%) on CUB-200 (Sec. 4.6).

## 2 Related Work

**Example-based explanations on the distinction task**   The *distinction* task (Fig. 1) was introduced in prior studies Nguyen et al. (2021); Kim et al. (2022); Fel et al. (2023); Colin et al. (2022) to test the utility of an explanation method. Yet, many works Nguyen et al. (2021); Taesiri et al. (2022); Kim et al. (2022); Kenny et al. (2023); Jeyakumar et al. (2020); Chen et al. (2023b); Nguyen et al. (2024) showed users examples from *only the top-1* class, potentially limiting the utility of nearest neighbors and user accuracy. We find that, given the same budget of five examples, human distinction accuracy substantially improves if each example comes from a unique class among the top-5 predicted classes (Sec. 4.6). To our knowledge, our work is the first to report the benefit of class-wise contrastive examples to users' decision-making.

**Re-ranking for image classification**   Re-ranking is common in image retrieval Phan & Nguyen (2022); Zhang et al. (2020); Li et al. (2023). For image classification, re-ranking the nearest neighbors of a $k$-NN image classifier can improve its classification accuracy Taesiri et al. (2022). Yet, here, we re-rank the top-$K$ predicted labels originally given by a classifier **C**.

**Ensembles for image classification**   Our method for combining two separate models (**C** and **S**) adds to the long literature of model ensembling. Specifically, our method of sampling hard, negative pairs of images using **C**'s predicted labels to train the image comparator **S** is akin to *boosting* Schapire (2003). That is, **S** is trained with the information of which negative pairs are considered "hard" (i.e., very similar images) according to **C**—the approach we find more effective than random sampling (Appendix B.4). While boosting aims to train a set of classifiers for the exact same task, our **C** is a standard many-way image classifier and **S** is a binary classifier that compares two input images.

Our combination of **C** and **S** is also inspired by PoE Hinton (1999). Unlike the traditional PoE algorithms, which train both experts at the same time and for the same task, here **C** and **S** are two separate classifiers with different input and output structures. **A key difference** from standard PoE and boosting techniques is that we leverage training-set examples (PCNN) at the test time of **S**, improving **C** × **S** model accuracy further over the baseline **C**. Also, our model does not strictly follow the PoE framework's requirement of conditional independence between experts because the confidence scores that image compartor **S** assigns to most-probable classes can be influenced by the initial ranking from **C**.

**Prototype-based image classifiers**   Many prototype-based classifiers Chen et al. (2019); Taesiri et al. (2022) operate at the patch level. Instead, our classifier operates at the image level, similar to $k$-NN classifiers Nguyen et al. (2021). While most prior prototype-based classifiers are single models, we combine two models (**C** and **S**) into one. Furthermore, the scores given by **S** enable an interpretation of how original predictions are re-ranked (Fig. 2).

## 3 Methods

### 3.1 Tasks

**Task 1: Single-label, many-way image classification**   Let **C** be a frozen, pretrained image classifier that takes in an image $x$ and outputs a softmax probability distribution over all $c$ possible classes, e.g., $c = 200$ for CUB-200 Wah et al. (2011). Let **S** be an image comparator that takes in two images and outputs a sigmoid score predicting whether they belong to the same class (Fig. 3). Our goal is to improve the final classification accuracy without changing **C** by leveraging a separate image comparator network and PCNN (Fig. 2).

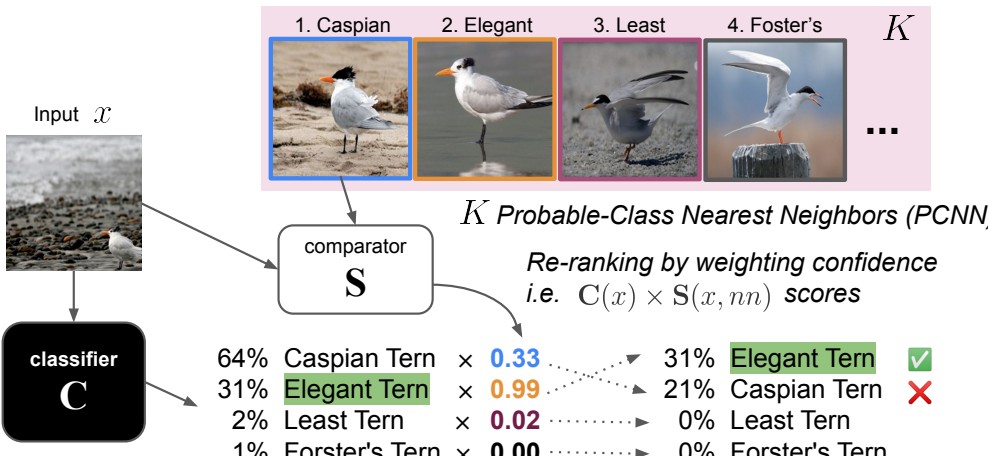

Figure 2: $\mathbf{C} \times \mathbf{S}$ re-ranking algorithm: From each class among the top-$K$ predicted classes by $\mathbf{C}$, we find the nearest neighbor $nn$ to the query $x$ and compute a sigmoid similarity score $\mathbf{S}(x, nn)$, which weights the original $\mathbf{C}(x)$ probabilities, re-ranking the labels. See Algorithm 2 for the written algorithm.

**Task 2: Distinction task for humans**   Following Nguyen et al. (2021); Taesiri et al. (2022), we provide each user with the input query, the top-1 prediction given by $\mathbf{C}$, and an explanation (e.g., five PCNN images; Fig. 8) and ask the user to accept or reject the top-1 predicted label.

### 3.2   Datasets and pretrained classifiers C

We train and test our method on three standard fine-grained image classification datasets of birds, cars, and dogs. To study the generalization of our findings, we test a total of 10 classifiers $\mathbf{C}$ (4 bird, 3 car, and 3 dog classifiers) of varying architectures and accuracy.

**CUB-200** (CUB-200-2011) Wah et al. (2011) has 200 bird species, with 5,994 images for training and 5,794 for testing (samples in Fig. 5a). We test four different classifiers: a ResNet-50 pretrained on iNaturalist Van Horn et al. (2018) and finetuned on CUB-200 (85.83% accuracy) by Taesiri et al. (2022); and three ImageNet-pretrained ResNets (18, 34, and 50 layers) finetuned on CUB-200 with 60.22%, 62.81%, and 62.98% accuracy, respectively.

**Cars-196** (Stanford Cars) Krause et al. (2013) includes 196 distinct classes, with 8,144 images for training and 8,041 for testing (samples in Fig. 5b). We use ResNet-18, ResNet-34, and ResNet-50, all pretrained on ImageNet and then finetuned on Cars-196. Their top-1 accuracy scores are 86.17%, 82.99%, and 89.73%, respectively.

**Dogs-120** (Stanford Dogs) Khosla et al. (2011) has a total 120 of dog breeds, with 12,000 images for training and 8,580 images for testing (samples in Fig. 5c). We test three models: ResNet-18, ResNet-34, and ResNet-50, all pretrained on ImageNet and then finetuned on Dogs-120, achieving top-1 accuracy of 78.75%, 82.58%, and 85.82%.

### 3.3   Re-ranking using both image comparator S and classifier C

**PCNN**   is a set of $K$ *nearest*-neighbor images to the query where each image is taken from one training-set class among the top-$K$ predicted classes by $\mathbf{C}$ (see Fig. 2). We empirically test $K = \{1, 2, 3, 5, 10, 15\}$ in Appendix B.3 and find $K = 10$ to be optimal.

The **distance metric** for finding nearest neighbors per class is $L_2$ (using `faiss` framework) Johnson et al. (2019) at the average pooling of the last `conv` features of $\mathbf{C}$.

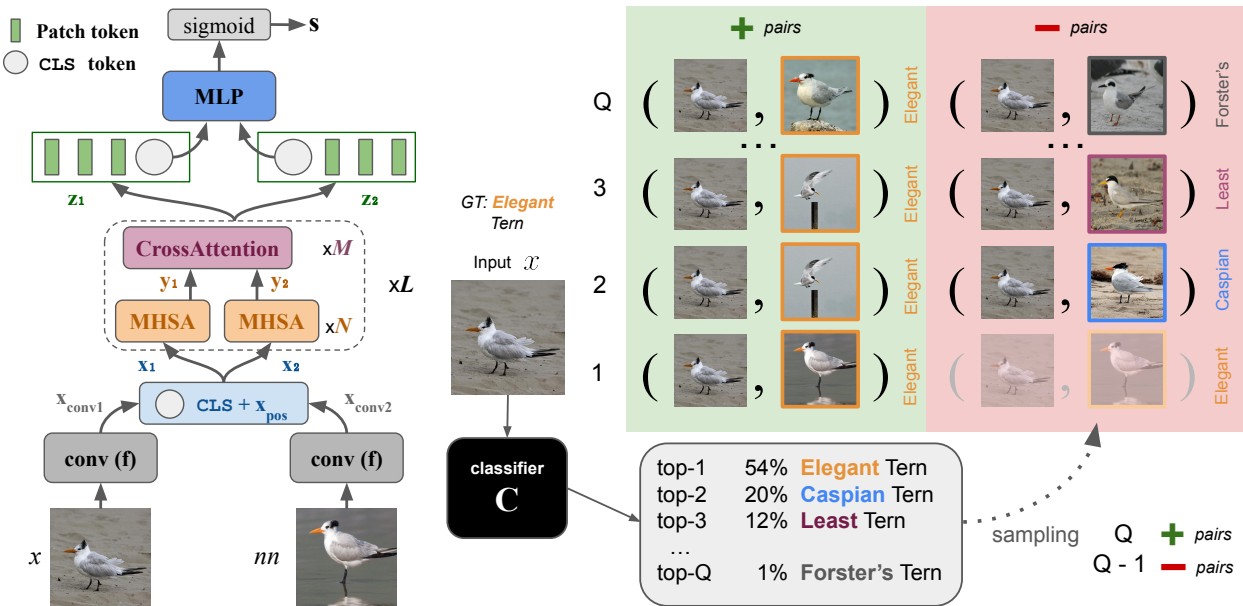

Figure 3: Our comparator takes in a pair of images $(x, nn)$ and outputs a sigmoid score $\mathbf{s} = \mathbf{S}(x, nn) \in [0, 1]$ indicating whether two images belong to the same class. $L$, $M$, and $N$ are the depths of the respective blocks.

Figure 4: For each training-set image $x$, we sample $Q$ nearest images from the groundtruth class of $x$ to form $Q$ positive pairs $\{(x, nn_+^i)\}_{Q}^{i=1}$. To sample $Q$ hard, negative pairs: Per non-groundtruth class among the top-$Q$ predicted classes from $\mathbf{C}(x)$, we take the nearest image to the input. Here, when the groundtruth label (Elegant Tern) is among the top-$Q$ labels, there would be only $Q - 1$ negative pairs.

**Re-ranking algorithm**  Given a well-trained comparator $\mathbf{S}$ and the obtained PCNN, we repeat the following for each class among the top-$K$ classes: Multiply each original confidence score in $\mathbf{C}(x)$ by a corresponding score $\mathbf{S}(x, nn)$ where $nn$ is the nearest neighbor from a corresponding predicted class (see Fig. 2).

Based on the newly weighted scores $\mathbf{C}(x) \times \mathbf{S}(x, nn)$, we re-rank the top-$K$ labels. See Fig. 5 for examples of how re-ranking corrects originally wrong predictions by $\mathbf{C}$.

### 3.4  The architecture and training of comparator S

#### 3.4.1  Network architecture

Our image comparator $\mathbf{S}$ follows closely the design of the CrossViT (Chen et al., 2021), which takes in a pair of images (see Fig. 3). The image patch embeddings are initialized with convolutional features from a pretrained convolutional network (here, we directly use the classifier $\mathbf{C}$ for convenience but such coupling is not mandatory; see Sec. 4.4).

**Details**  $\mathbf{f}$ is the subnetwork up to and including the last conv layer of a given classifier $\mathbf{C}$. For example, if we train a comparator $\mathbf{S}$ for ResNet-50, $\mathbf{f}$ would be layer4 per PyTorch definition or conv5_x in He et al. (2016). Using the pretrained feature extractor $\mathbf{f}$, we extract two conv embeddings $\mathbf{x}_{conv1}$ and $\mathbf{x}_{conv2}$ $\in \mathbb{R}^{D \times H \times W}$ where $H \times W$ are spatial dimensions and $D$ is the depth. We then flatten and transpose these embeddings to shape them into $\mathbb{R}^{T \times D}$ (e.g., $T = H \times W = 49$ and $D = 2048$ for ResNet-50). These embeddings are subsequently prepended with the CLS token $\mathbf{x}_{cls} \in \mathbb{R}^{1 \times D}$ and added to the learned positional embedding $\mathbf{x}_{pos} \in \mathbb{R}^{(1+T) \times D}$. After that, the two image embeddings $\mathbf{x}_1$ and $\mathbf{x}_2 \in \mathbb{R}^{(1+T) \times D}$ are fed into the first Transformer block that consists of $N$ MHSA layers and $M$ CrossAttention layers (Fig. 3). We repeat the Transformer block $L$ times.

**MHSA** is the multi-headed self-attention operator Vaswani et al. (2017) and **Cross Attention** refers to the Cross-Attention token fusion block in CrossViT Chen et al. (2021). These two blocks produce $\mathbf{y}_1$, $\mathbf{y}_2$, and $\mathbf{z}_1$, $\mathbf{z}_2 \in \mathbb{R}^{(1+T) \times D}$, respectively (Fig. 3). Next, we concatenate the two CLS tokens from $\mathbf{z}_1$ and $\mathbf{z}_2$ as

input for a 4-layer MLP. Each layer uses GeLU Hendrycks & Gimpel (2016) and Batch Normalization Ioffe & Szegedy (2015), except the last layer. The output of the MLP is a sigmoid value $\mathbf{s}$ that indicates whether $x$ and $nn$ belongs to the same class. Full architecture details are in Appendix A.

### 3.4.2 Training the comparator S

**Objectives**   We aim to train $\mathbf{S}$ to separate image pairs taken in the same class from those pairs where images are from two different classes. As standard in contrastive learning Chen et al. (2020), we first construct a set of positive pairs and a set of negative pairs from the training set, and then train $\mathbf{S}$ using a binary sigmoid cross-entropy loss. Note that training the comparator also finetunes the pretrained conv layers $\mathbf{f}$, which are part of the comparator model (Fig. 3).

**Augmentation**   Because the three tested fine-grained classification datasets are of smaller sizes, we apply TrivialAugment (Müller & Hutter, 2021) to image pairs during training to reduce overfitting. For image pre-processing, we first resize images so that their smaller dimension is 256, and then take a center crop of $224 \times 224$ from the resized image.

**Optimization**   For all three datasets, we train comparators $\mathbf{S}$ for 100 epochs with a batch size of 256 using Stochastic Gradient Descent (SGD) optimizer with a 0.9 momentum and the OneCycleLR (Smith & Topin, 2019) learning-rate scheduler.

Some hyperparameters including learning rates vary depending on the datasets. See Appendix A for more training details including optimization hyperparameters, data augmentation, and training loss.

### 3.4.3 Sampling positive and negative pairs

For each training-set example $x$, we construct a set of positive pairs $\{(x, nn_+)\}$ and negative pairs $\{(x, nn_-)\}$ (Fig. 4). To find *nearest* images, we use the distance metric described in Sec. 3.3.

**positive pairs**   We take $Q$ nearest images $nn_+$ to the query $x$ from the same class of $x$ (e.g., Elegant Tern in Fig. 4).

**negative pairs**   One can also take $nn_-$ nearest images from the random non-groundtruth classes. However, in the preliminary experiments, we find that taking $nn_-$ from a random class (e.g., among 200 bird classes) produces "easy" negative pairs $(x, nn_-)$, i.e., images are often too visually different, not strongly encouraging $\mathbf{S}$ to learn to focus on subtle differences between fine-grained species as effectively as our "hard" negatives sampling.

**Sampling using classifier C**   First, we observe that pretrained classifiers $\mathbf{C}$ often have a very **high top-10 accuracy** (e.g., 98.63% on CUB-200; Appendix G) and therefore tend to place species visually similar to the ground-truth class among the top-$Q$ labels. Therefore, we leverage the predictions $\mathbf{C}(x)$ of the classifier $\mathbf{C}$ on $x$ to sample hard negatives. That is, we sample $Q$ $nn_-$ nearest images to the query. Yet, each $nn_-$ is from a class among the top-$Q$ predicted labels for $x$, i.e., from the $\mathbf{C}(x)$ (see example pairs in Appendix I.3). As illustrated in Fig. 4, if the groundtruth class appears in the top-$Q$ labels, we exclude that corresponding negative pair, arriving at $Q - 1$ negative pairs. In this case, we will remove one positive pair to make the data balance. In sum, if the groundtruth class is in the top-$Q$ labels, we will produce $Q$ positive and $Q - 1$ negative pairs. Otherwise, we would produce $Q$ positive and $Q$ negative pairs.

Empirically, we try $Q \in \{3, 5, 10, 15\}$ and find $Q = 10$ to yield the best comparator based on its test-set binary-classification accuracy (see the results of tuning $Q$ in Appendix B.2). See Appendix B for further experiments supporting our design choices.

## 4   Results

In this section, we demonstrate that PCNN examples enhance both AI and human accuracy. First, we use PCNN examples to train an image comparator $\mathbf{S}$, which improves classifier $\mathbf{C}$'s predictions via the re-ranking algorithm described in Sec. 3.3. Second, when shown PCNN examples, human users increase their accuracy in distinguishing correct from incorrect predictions by nearly $+10$ points.

### 4.1 $\mathbf{C} \times \mathbf{S}$ re-ranking consistently outperforms classifier C

Here, we aim to test how our re-ranking algorithm (Sec. 3.3) improves upon the original classifiers $\mathbf{C}$.

**Experiment** For each of the 10 classifiers listed in Sec. 3.2, we train a corresponding comparator $\mathbf{S}$ (following the procedure described in Sec. 3.4) and form a $\mathbf{C} \times \mathbf{S}$ model.

Table 1: On all three ResNet (RN) architectures and three datasets, our $\mathbf{C} \times \mathbf{S}$ consistently improves the top-1 classification accuracy (%) over the original classifiers $\mathbf{C}$ (e.g., by +11.48 on CUB-200) and also a baseline re-ranking $\mathbf{C} \rightarrow \mathbf{S}$ (which uses only $\mathbf{S}$ scores in re-ranking). "Pretraining" column specifies the datasets that $\mathbf{C}$ models were pretrained (before fine-tuning on the target dataset).

| Classifier architecture | | ResNet-18 (a) | | | ResNet-34 (b) | | | ResNet-50 (c) | | |
|---|---|---|---|---|---|---|---|---|---|---|
| Dataset | Pretraining | $\mathbf{C}$ | $\mathbf{C} \rightarrow \mathbf{S}$ | $\mathbf{C} \times \mathbf{S}$ | $\mathbf{C}$ | $\mathbf{C} \rightarrow \mathbf{S}$ | $\mathbf{C} \times \mathbf{S}$ | $\mathbf{C}$ | $\mathbf{C} \rightarrow \mathbf{S}$ | $\mathbf{C} \times \mathbf{S}$ |
| CUB-200 | iNaturalist | n/a | n/a | n/a | n/a | n/a | n/a | 85.83 | 87.72 | 88.59 (+2.76) |
| | ImageNet | 60.22 | 66.78 | 71.09 (+10.87) | 62.81 | 71.92 | 74.59 (+11.78) | 62.98 | 71.63 | 74.46 (+11.48) |
| Cars-196 | ImageNet | 86.17 | 85.70 | 88.27 (+2.10) | 82.99 | 83.57 | 86.02 (+3.03) | 89.73 | 89.90 | 91.06 (+1.33) |
| Dogs-120 | ImageNet | 78.75 | 75.34 | 79.58 (+0.83) | 82.58 | 80.82 | 83.62 (+1.04) | 85.82 | 83.39 | 86.31 (+0.49) |

**Results** Our $\mathbf{C} \times \mathbf{S}$ models outperform classifiers $\mathbf{C}$ consistently across all three architectures (ResNet-18, ResNet-34, and ResNet-50) and all three datasets (see Tab. 1). The largest gains on CUB-200, Cars-196, and Dogs-120 are +11.78, +3.03, and +1.04 percentage points (pp), respectively. That is, our method works best on bird images, followed by cars and dogs.

A trend (see Tab. 1 and Fig. 6) is that when the original classifier $\mathbf{C}$ is weaker, our re-ranking often yields a larger gain. Intuitively, a weaker classifier's predictions benefit more from revising based on extra evidence (PCNN) and an external model (comparator $\mathbf{S}$). Yet, on CUB-200, we also improve upon the best model (iNaturalist-pretrained ResNet-50) by +2.76 (85.83% → 88.59%). Note that while the gains on Dogs-120 are modest (Tab. 1), dog images are the noisiest among the three tested image types, and therefore the small but consistent gains on Dogs-120 are encouraging.

See Fig. 5 for examples of how our $\mathbf{C} \times \mathbf{S}$ model corrects originally wrong predictions made by ResNet-50 (from Indigo Bunting → Green Jay on CUB-200, from Aston Martin → Ferrari FF on Cars-196, and from Irish Terrier → Otterhound). More qualitative examples are in Appendix I.1.

### 4.2 Hyperparameter tuning and ablation studies

We perform thorough tests over many choices for each hyperparameter and present the main findings below.

**At test time, comparing input with multiple nearest neighbors yields slightly better weights for re-ranking** Our proposed re-ranking (Fig. 2) takes $n = 1$ image per class among the top-$K$ classes to compute a similarity score indicating how well the input matches a class. However, would one nearest image be sufficient to represent an entire class?

We find that for each class among the top-$K$, increasing from $n = 1$ to 2 or 3 then taking the average similarity scores over the $n$ image pairs only *marginally* increase the $\mathbf{C} \times \mathbf{S}$ accuracy (from 88.59% → 88.83% on CUB-200 and 91.09% → 91.20% on Cars-196; full results in Appendix B.1). However, these marginal gains require querying $\mathbf{S}$ two to three times more, entailing a much slower run time. Therefore, we propose to use $n = 1$ nearest neighbor per class.

**Sampling image pairs from the top-$Q$ classes where $Q = 10$** A key to training our comparator (Fig. 4) is to sample image pairs from the top-$Q$ labels predicted by $\mathbf{C}$. We test training $\mathbf{S}$ using different values of $Q = 3, 5, 10, 15$ and find that the comparator's accuracy starts to saturate at $Q = 10$ (Appendix B.2). Hence, we use $Q = 10$ in all experiments.

**Hard negatives are more useful than easy, random negatives in training S** We find that training $\mathbf{S}$ using negative pairs constructed using the nearest $nn_-$ from *random* classes among the 200 CUB classes yields a RN50 × $\mathbf{S}$ accuracy of 86.55%. In contrast, if we sample negative pairs from the top-$Q$ predicted classes returned by $\mathbf{C}$, the accuracy substantially increases by +2.04 pp to 88.59% (as reported in Tab. 2).

In another experiment, we repeat our hard-negative sampling from the top-$Q$ classes *but* in each class, we take the 2nd or 3rd image (instead of the 1st) nearest to the query image to be the class-representative image. Yet, the $\mathbf{C} \times \mathbf{S}$ accuracy decreases from 88.59% to 88.06% and 88.21%, respectively. In sum, both experiments show that sampling negative pairs using the 1st nearest neighbors from probable classes yields substantially better comparators than from random classes (Appendix B.4).

**Re-ranking using the $\mathbf{C} \times \mathbf{S}$ product is better than using the scores of $\mathbf{S}$ alone**  To understand the importance of the product of two scores $\mathbf{C}(x) \times \mathbf{S}(x, nn)$ in re-ranking (Fig. 2), we test ablating $\mathbf{C}(x)$ away from this product, using only $\mathbf{S}(x, nn)$ scores to re-rank the top-predicted labels. We refer to this approach as $\mathbf{C} \to \mathbf{S}$.

We find $\mathbf{C} \to \mathbf{S}$ re-ranking to sometimes even hurt the accuracy compared to the original $\mathbf{C}$ (e.g., 78.75% → 75.34% on Dogs-120; Tab. 1). Yet, with the same comparator $\mathbf{S}$, using the product of two scores $\mathbf{C} \times \mathbf{S}$ consistently outperforms not only the original classifier $\mathbf{C}$ but also this $\mathbf{C} \to \mathbf{S}$ baseline re-ranking by an average gain of +3.27, +2.06, and +3.32 pp on CUB-200, Cars-196, and Dogs-120, respectively (Tab. 1).

**TrivialAugment improves generalization of $\mathbf{S}$**  Over all three datasets, when training comparators using only training-set images (no augmentation), the training-validation loss curves (full results in Appendix B.8) show that the models overfit.

To combat this issue, besides early stopping, we incorporate TrivialAugment Müller & Hutter (2021) into the training and find it to help the learning to converge faster and reach better accuracy. Compared to

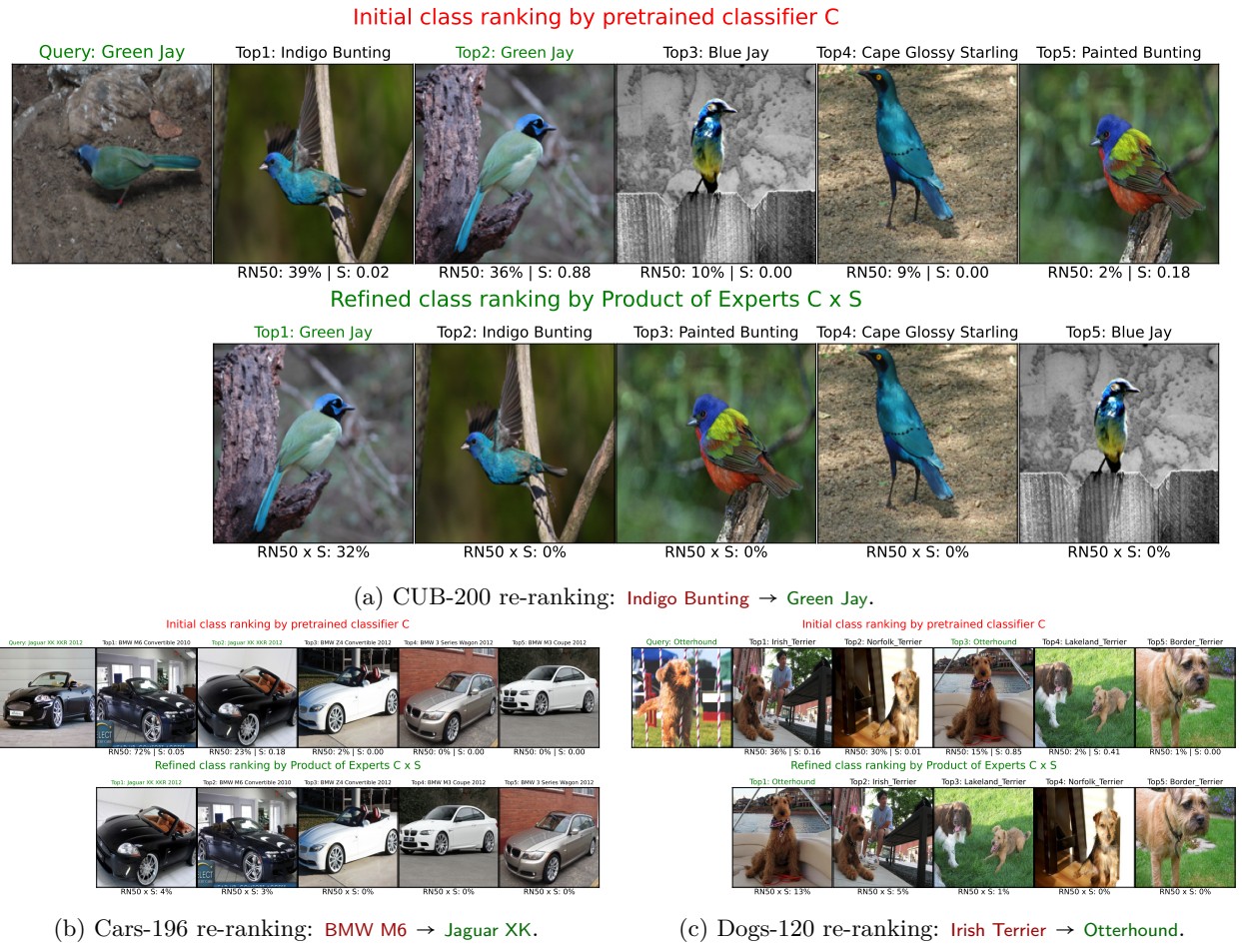

(a) CUB-200 re-ranking: Indigo Bunting → Green Jay.

(b) Cars-196 re-ranking: BMW M6 → Jaguar XK.

(c) Dogs-120 re-ranking: Irish Terrier → Otterhound.

Figure 5: $\mathbf{C} \times \mathbf{S}$ model successfully corrects originally wrong predictions made by ResNet-50.

using no data augmentation, TrivialAugment leads to test-set binary-classification accuracy gains of +1.32 on CUB-200, +1.52 on Cars-196, and +1.21 on Dogs-120.

### 4.3 Training comparators on image pairs is key

At the core of our method is the image comparator $\mathbf{S}$. Yet, one might wonder: (1) Is training $\mathbf{S}$ necessary? Would using cosine similarity in a common feature space be sufficient? and (2) How important are the components (self-attention, cross-attention, MLP) of the comparator?

**Both cosine and Earth Mover's distance in a fine-grained image classifier's feature space poorly separate visually similar species. Contrastively training comparators is a key.** First, we test using cosine similarity in the pretrained feature space of a classifier Taesiri et al. (2022) of 85.83% accuracy on CUB-200. That is, we use the avgpool features of ResNet-50 (i.e., average pooling after the last conv layer) to be our image embeddings. However, we find that $\mathbf{C} \to \mathbf{S}$ re-ranking using this similarity metric leads to poor CUB-200 accuracy compared to using our trained comparator (60.20% vs. 87.72%).

We also test re-ranking using patch-wise similarity by following Phan & Nguyen (2022); Zhang et al. (2020) and using the Earth Mover's distance (EMD) at the 49-patch embedding space ($7 \times 7 \times 2048$) of layer4. Yet, this patch-wise image similarity also leads to poor accuracy (54.83%). In sum, we find the pretrained conv features of a state-of-the-art classifier are not sufficiently discriminative to separate probable classes (Fig. 2).

In the next experiment, we find that training even a simple MLP contrastively on the image pairs sampled following our $\mathbf{C}$-based sampling (Sec. 3.4.3) dramatically improves the re-ranking to 83.76% on CUB-200.

**MLP in the comparator S plays the major role and adding Transformer blocks slightly improves accuracy further** We perform an ablation study to understand the importance of the four main components— finetuned features $\mathbf{f}$, MHSA, cross attention, and MLP—of the proposed comparator architecture (Fig. 3).

First, we find that finetuning $\mathbf{f}$ while training the comparator leads to better binary classification accuracy (94.44% vs. 91.94%; see Appendix B.6). Second, we find that training the 4-layer MLP with the pretrained features $\mathbf{f}$ (i.e., $L = 0$, no MSHA and no cross attention) leads to a comparator that has a $\mathbf{C} \to \mathbf{S}$ re-ranking accuracy of 83.76% on CUB-200. Adding MSHA and cross-attention blocks led to the best comparator, pushing this number by +3.55 to 87.31%. More details are in Appendix B.9.

**Sanity checks** We perform additional sanity checks to confirm our trained binary classifier works as expected. Indeed, $\mathbf{S}$ outputs Same class 100% of the time when two input images are identical and nearly 0% of the time given a pair of two random images (that are likely from two different classes) (Appendix F). Another insight is that the comparator tends to assign higher mean confidence scores when it is correct compared to when it is wrong (Appendix F.2).

### 4.4 Well-trained comparator S works well with an arbitrary classifier C in a C × S model

Sec. 4.1 shows that using a given comparator $\mathbf{S}_1$ trained on data sampled based on a specific classifier $\mathbf{C}_1$, one can form a $\mathbf{C}_1 \times \mathbf{S}_1$ model that outperforms the original $\mathbf{C}_1$.

Here, we test if such coupling is necessary: Would the same comparator $\mathbf{S}_1$ form a high-performing $\mathbf{C} \times \mathbf{S}$ with an *unseen*, black-box classifier $\mathbf{C}_2$?

**Experiment** First, we take an image comparator $\mathbf{S}_1$ from Sec. 4.1 that was trained on data sampled based on a high-performing CUB ResNet-50 classifier by Taesiri et al. (2022) (85.83% accuracy; Tab. 1c). Then, in the $\mathbf{C} \times \mathbf{S}$ setup (Sec. 3.3), we test using the same $\mathbf{S}_1$ to re-rank the predictions of *five* different CUB classifiers: three different ResNets (Tab. 1), a ViT/B-16 (82.40%), and an NTS-Net (87.04%) Yang et al. (2018).

**Results** Interestingly, $\mathbf{S}_1$ works well with many unseen, black-box classifiers in a $\mathbf{C} \times \mathbf{S}$ model, consistently increasing the accuracy over the classifiers alone. Across unseen classifiers, we witness significant accuracy boosts (Fig. 6; --) of around +20 pp for all ResNet models, which were pretrained on ImageNet and finetuned on CUB-200. With ViT and NTS-Net, the gains are smaller, yet consistent. Detailed quantitative numbers are reported in Appendix C.

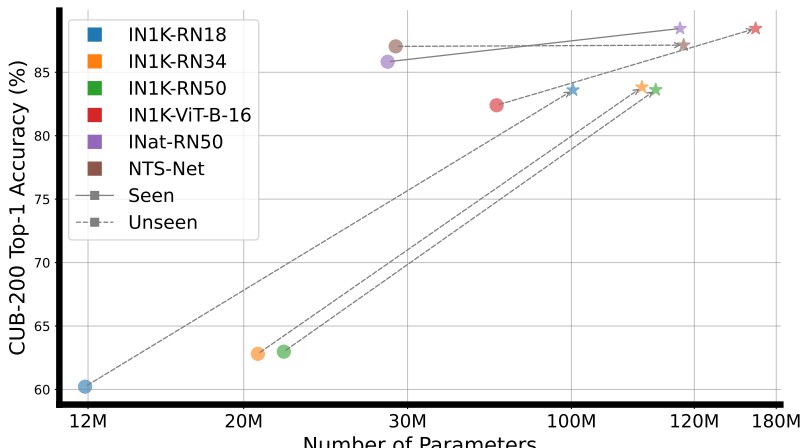

Figure 6: When pairing a well-trained comparator $\mathbf{S}_1$ with an unseen, black-box classifier ($\bullet$), our $\mathbf{C} \times \mathbf{S}$ models ($\star$) consistently yields a higher accuracy on CUB-200. Along the x-axis, each $\star$ shows the total number of parameters of $\mathbf{S}_1$ and a paired classifier.

### 4.5  $\mathbf{C} \times \mathbf{S}$ outperforms prototype-based classifiers

Our PCNN-based approach is related to many **explainable methods** that compares the input to examplar images Papernot & McDaniel (2018), prototypical patches Chen et al. (2019) or both images and patches Taesiri et al. (2022). Here, we aim to compare our method with these state-of-the-art models on CUB-200 (Tab. 2), Cars-196 (Tab. 3), and Dogs-120 (Tab. 4).

**k-Nearest Neighbors ($k$-NN)** operates by comparing an input image against *the entire* training set at the image level. In contrast, our $\mathbf{C} \times \mathbf{S}$ uses only images from $K = 10$ classes out of all classes (e.g. 200 for CUB) for re-ranking. Following Taesiri et al. (2022), we use $k = 20$ and implement a baseline ($k$-NN + $\mathbf{S}$) that compares images using cosine similarity in the pretrained ResNet-50's avgpool features (Tab. 2; 85.46% accuracy). To understand the importance of our re-ranking, we also test another $k$-NN baseline that uses our comparator's scores instead of cosine similarity.

**Consistent results across all three datasets**  First, we find that using $\mathbf{S}$ as the distance function for $k$-NN results in competitive accuracy on all three datasets, specifically, 86.88% on CUB-200 (Tab. 2) and 88.90% on Cars-196 (Tab. 3). This $k$-NN + $\mathbf{S}$ result shows that **S is a strong generic image comparator**. That is, here, $\mathbf{S}$ can be used to compare training-set images that are even outside the distribution of top-$Q$ classes that it was trained on.

Second, our $\mathbf{C} \times \mathbf{S}$ outperforms both $k$-NN + cosine and $k$-NN + $\mathbf{S}$ baselines consistently by around +1 to +3 pp, suggesting that the $\mathbf{S}$-based re-ranking method is effective.

Third, our $\mathbf{C} \times \mathbf{S}$ method outperforms the state-of-the-art explainable re-ranking methods of CHM-Corr and EMD-Corr Taesiri et al. (2022) by a large margin of around +4 on CUB-200 and Cars-196 (Tabs. 2 and 3) and +0.73 on Dogs-120 (Tab. 4). While both techniques are 2-stage classification, the major technical difference is that our $\mathbf{C} \times \mathbf{S}$ leverages a high-performing black-box $\mathbf{C}$ and further finetune its top-$K$ predicted labels using PCNNs. In contrast, CHM-Corr and EMD-Corr relies on a slow and often lower-performing $k$-NN to first rank the training-set images, and then re-rank the shortlisted images in their 2nd stage.

Fourth, by a large margin, our $\mathbf{C} \times \mathbf{S}$ outperforms all models in the prototypical-part-based family, which learn representative patch prototypes for each class and compare them to the input image at test time. From the explainability viewpoint, $\mathbf{C} \times \mathbf{S}$ reveals insights into how labels are re-ranked but not how they are initially predicted by the black-box $\mathbf{C}$. In contrast, prototype-based family aims to show insights into how the similarity between input patches and prototypes contributes to initial image-label predictions.

Table 2: **CUB-200** top-1 test-set accuracy (%). All models are finetuned on CUB-200 from an iNaturalist-pretrained ResNet-50 backbone. For our method, we report the mean and std over 3 random seeds (see Sec. B.7). Prototypical part-based classifiers typically use full, uncropped images, and 10 part prototypes per class. Accuracy $^\dagger$ is from Wang et al. (2023b).

**E**: Using training-set **E**xamples at *test* time to make predictions.
**I**: Comparing images at the **I**mage level.
**P**: Comparing images at the **P**atch level.
**R**: **R**e-ranking initial predictions of a classifier.

| Classifier | Ex | Img | Patch | R | Acc |
|---|---|---|---|---|---|
| $k$-NN + cosine Taesiri et al. (2022) | ✓ | ✓ | - | - | 85.46 |
| $k$-NN + **S** | ✓ | ✓ | - | - | 86.88 |
| ProtoPNet Chen et al. (2019) | - | - | ✓ | - | 81.10$^\dagger$ |
| PIPNet Nauta et al. (2023) | - | - | ✓ | - | 82.00 |
| ProtoTree Nauta et al. (2021) | - | - | ✓ | - | 82.20 |
| ProtoPool Rymarczyk et al. (2022) | - | - | ✓ | - | 85.50 |
| Def-ProtoPNet Donnelly et al. (2021) | - | - | ✓ | - | 86.40 |
| TesNet Wang et al. (2021) | - | - | ✓ | - | 86.50$^\dagger$ |
| ST-ProtoPNet Wang et al. (2023b) | - | - | ✓ | - | 86.60 |
| ProtoKNN Ukai et al. (2023) | ✓ | - | ✓ | - | 87.00 |
| CHM-Corr Taesiri et al. (2022) | ✓ | ✓ | ✓ | ✓ | 83.27 |
| EMD-Corr Taesiri et al. (2022) | ✓ | ✓ | ✓ | ✓ | 84.98 |
| **C** × **S** (ours) | ✓ | ✓ | - | ✓ | **88.59** ± 0.17 |

Table 3: **Cars-196** top-1 test-set accuracy (%). All models are finetuned on Cars-196 from an ImageNet-pretrained ResNet-50 backbone. Accuracy$^\dagger$ of ProtoPNet is from Keswani et al. (2022) & accuracy$^*$ of ProtoPShare Rymarczyk et al. (2021) is from Nauta et al. (2023).

| Classifier | Ex | Img | Patch | R | Acc |
|---|---|---|---|---|---|
| $k$-NN + cosine | ✓ | ✓ | - | - | 87.48 |
| $k$-NN + **S** | ✓ | ✓ | - | - | 88.90 |
| ProtoPNet | - | - | ✓ | - | 85.31$^\dagger$ |
| ProtoPShare | - | - | ✓ | - | 86.40$^*$ |
| PIPNet | - | - | ✓ | - | 86.50 |
| ProtoTree | - | - | ✓ | - | 86.60 |
| ProtoPool | - | - | ✓ | - | 88.90 |
| ProtoKNN | ✓ | - | ✓ | - | 90.20 |
| CHM-Corr | ✓ | ✓ | ✓ | ✓ | 85.03 |
| EMD-Corr | ✓ | ✓ | ✓ | ✓ | 87.40 |
| **C** × **S** (ours) | ✓ | ✓ | - | ✓ | **91.06** ± 0.15 |

Table 4: **Dogs-120** top-1 test-set accuracy (%). All models are finetuned on Dogs-120 from an ImageNet-pretrained ResNet-50 backbone. Accuracy$^\dagger$ is from Wang et al. (2023b) and MGProto is from Wang et al. (2023a).

| Classifier | Ex | Img | Patch | R | Acc |
|---|---|---|---|---|---|
| $k$-NN + cosine | ✓ | ✓ | - | - | 85.56 |
| $k$-NN + **S** | ✓ | ✓ | - | - | 82.33 |
| ProtoPNet | - | - | ✓ | - | 76.40$^\dagger$ |
| TesNet | - | - | ✓ | - | 82.40$^\dagger$ |
| Def-ProtoPNet | - | - | ✓ | - | 82.20$^\dagger$ |
| ST-ProtoPNet | - | - | ✓ | - | 84.00 |
| MGProto | - | - | ✓ | - | 85.40 |
| CHM-Corr | ✓ | ✓ | ✓ | ✓ | 85.59 |
| EMD-Corr | ✓ | ✓ | ✓ | ✓ | 85.57 |
| **C** × **S** (ours) | ✓ | ✓ | - | ✓ | **86.31** ± 0.03 |

### 4.6 PCNN improves human accuracy in predicting AI misclassifications on bird images

Given the effectiveness of PCNN examples in re-ranking, we are motivated to test them on human users. Specifically, we compare user' accuracy in the *distinction* task Kim et al. (2022) (i.e., telling whether a given classifier **C** is correct or not) when presented with top-1 class examples as in prior work Nguyen et al. (2021); Taesiri et al. (2022) compared to when presented with PCNN (Fig. 8a vs. b).

**Experiment** We randomly sample 300 correctly classified and 300 misclassified images by the **C** × **S** model from the CUB-200 test set (88.59% accuracy; Tab. 2). From our institution, we recruit 33 lay users for the test with top-1 class examples and 27 users for the PCNN test. Per test, each participant is given 30

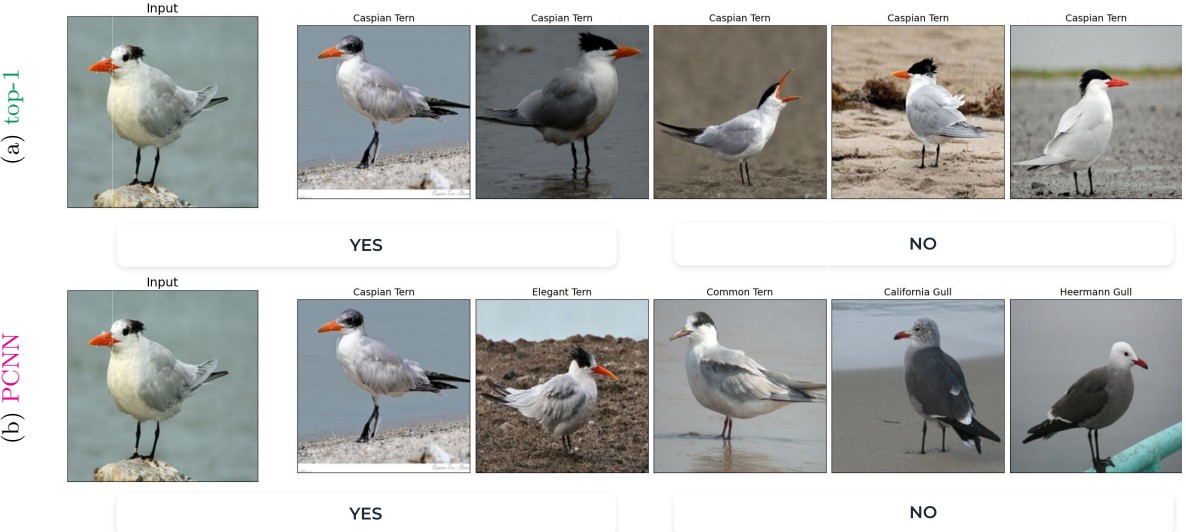

Figure 7: In both experiments, humans are asked whether the input image is Caspian Tern given that input, a model prediction, and either top-1 class examples (top) or PCNN explanations (bottom). When given only examples from the top-1 class, humans tend to accept the prediction, not knowing there are other very visually similar birds. Yet, the top-5 classes provide humans with a broader context which leads to better accuracy (64.58% vs. 54.55%; Sec. 4.6).

images, one at a time, and asked to predict (Yes or No) whether the top-1 predicted label is correct given the explanations (Fig. 7). To align with prior work, we choose $K = 5$ when implementing PCNN, i.e. we only show nearest examples from top-$K$ (where $K = 5$) classes to keep the explanations readable to users. More details in Appendix H.

**Results** We find that PCNN offers contrastive evidence for users to distinguish closely similar species, leading to better accuracy. That is, showing only examples from the top-1 class leads users to overly trust model predictions, rejecting only 22.28% of the cases where the AI misclassifies. In contrast, PCNN users correctly reject 49.31% of AI's misclassifications (Fig. 8b). Because users are given more contrastive evidence to make decisions, they tend to doubt AI decisions more often, resulting in lower accuracy when the AI is actually correct (79.78% vs. 90.99%; Fig. 8a).

Yet, on average, compared to showing only examples from the top-1 class in prior works Nguyen et al. (2021); Taesiri et al. (2022); Nguyen et al. (2024), PCNN improves user accuracy by a large margin of nearly +10 pp (54.55% → 64.58%). Our finding aligns with the literature that showing explanations helps users reduce over-reliance on machine predictions Buçinca et al. (2021); Schemmer et al. (2023); Chen et al. (2023a). We present extensive details of the human study in Tab. 17.

## 5 Limitations

While our re-ranking method shows a novel and exciting use of nearest neighbors, it comes with a major time-complexity limitation. That is, the re-ranking inherently adds extra inference time since we also need to query the comparator **S** $K$ times in addition to only calling the original classifier **C**. Empirically, when **C** is a ResNet-50 and $K = 10$, our **C** × **S** model is ~7.3× slower (64.55 seconds vs. 8.81 seconds for 1,000 queries). Our method runs also slower than $k$-NN and prototypical part-based classifiers Chen et al. (2019) but faster than CHM-Corr and EMD-Corr re-rankers Taesiri et al. (2022). For a detailed runtime comparison between other existing explainable classifiers and our method, along with our proposals to reduce runtime without sacrificing the classification accuracy, refer to Appendix E.

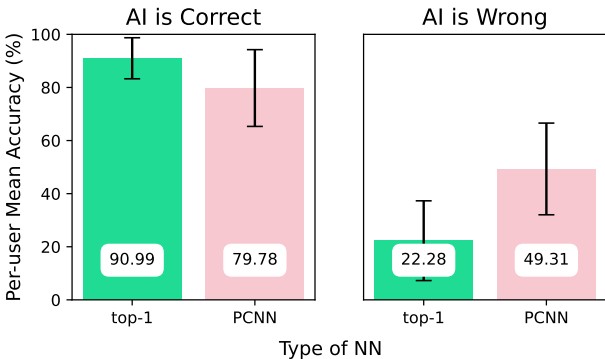

Figure 8: Users often accept when top-1 class neighbors are presented, leading to high accuracy when the AI is correct (a) and extremely poor accuracy when AI is wrong (b). PCNN ameliorates this limitation of the top-1 examples.

## 6    Discussion and Conclusion

In this work, we propose PCNN, a novel explanation consisting of $K$ images taken from the top-$K$ predicted classes to more fully represent the query. We first show that PCNN can be leveraged to improve the accuracy of a fine-grained image classification system without having to re-train the black-box classifier $\mathbf{C}$, which is increasingly a common scenario in this foundation model era. Second, we show that such a well-trained comparator $\mathbf{S}$ could be used with any arbitrary classifier $\mathbf{C}$. Our method also achieves state-of-the-art classification accuracy compared to existing prototype-based classifiers on CUB-200, Cars-196, and Dogs-120. Lastly, we find that showing PCNN also helps humans improve their decision-making accuracy compared to showing only top-1 class examples, which is a common practice in the literature.

An important aspect to consider is that while PCNN provides a comprehensive explanation by presenting multiple probable-class examples, it is crucial to ensure this does not reduce user confidence to a degree that negatively affects decision-making, especially in time-critical high-stakes scenarios. Balancing detailed information with user confidence is highly important for effective human-AI collaboration.

### Acknowledgement

We thank Son Nguyen and Hung Dao from KAIST, South Korea and Peijie Chen, Thang Pham, and Pooyan Rahmanzadehgervi from Auburn University for feedback on early results. AN was supported by the NSF Grant No. 1850117 & 2145767, and donations from NaphCare Foundation & Adobe Research. GN was supported by Auburn University Presidential Graduate Research Fellowship.

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

# A   Additional details of training, evaluation, and architecture

Unless otherwise stated, the same seed value of **42** was set to ensure reproducibility across Python, Numpy, and PyTorch operations. We employ the **faiss** (Johnson et al., 2019) library to index and retrieve nearest neighbors, utilizing the $L_2$ distance through the `IndexFlatL2` function.

## A.1   Training

Our experiments were conducted using two NVIDIA A100-SXM4-40GB GPUs, leveraging PyTorch version 1.13.0+cu117.

**CUB-200**   From the training set of 5,994 images of CUB-200, we follow the sampling procedure in Fig. 4 and collect a total of 113,886 pairs (where the positive pairs and negative pairs take $\approx 50\%$).

**Cars-196**   We repeat the same procedure but collect 154,736 pairs with the same positive/negative ratio from 8,144 images.

**Dogs-120**   We collect 228,000 pairs with the same positive/negative ratio from 12,000 dog samples.

In training an image comparator $\mathbf{S}$, given $B$ image pairs as training samples, and a true binary label $y_i$ (positive or negative) for each $i$-th sample pair, we train $\mathbf{S}$ to minimize the standard binary cross-entropy loss:

$$L = -\frac{1}{B} \sum_{i=1}^{B} y_i \log\left(\text{sigmoid}(o_i)\right) + (1 - y_i) \log\left(1 - \text{sigmoid}(o_i)\right) \tag{1}$$

To select the best training checkpoint for our image comparator $\mathbf{S}$ during training, we use the F1 score, which balances precision and recall, as our key metric. This process involves evaluating the model at each of the 100 training epochs. The F1 score helps us identify the checkpoint where the model best distinguishes between similar (positive) and dissimilar (negative) image pairs, ensuring a balance between accurately identifying true positives and minimizing false positives. This method aids in choosing a model that is not only accurate but also generalizes well, avoiding overfitting.

For CUB-200, we train $\mathbf{S}$ for 100 epochs with a batch size of 256 and a max learning rate of 0.01 for OneCycleLR Smith & Topin (2019) scheduler. The model architecture for this dataset is defined by $M = N = 4$ and $L = 2$. For Cars-196 and Dogs-120, we use the same batch size and number of epochs, but adjust the max learning rate to 0.1 and 0.01 for OneCycleLR scheduler, respectively. In both datasets, we use $M = N = 3$ and $L = 3$. Unless specified differently, both the self- and cross-attention Transformer blocks have 8 heads, and for the sampling process, $Q = 10$.

For the pretrained convolutional layers $\mathbf{f}$, we always use the last `conv` layer in ResNet architectures. $\mathbf{f}$ is the `layer4` per PyTorch definition or `conv5_x` in He et al. (2016). The `conv` features given by $\mathbf{f}$ are $\mathbf{x}_{\text{conv1}}$ and $\mathbf{x}_{\text{conv2}} \in \mathbb{R}^{2048 \times 7 \times 7}$) for ResNet-50 and $\mathbb{R}^{512 \times 7 \times 7}$) for both ResNet-34 and ResNet-18.

The MLPs consist of 4 layers, with each utilizing Gaussian Error Linear Units (GeLU) Hendrycks & Gimpel (2016) and Batch Normalization Ioffe & Szegedy (2015), except for the last two layers. See the definition of MLPs below:

**MLP architecture**

```
self.net = nn.Sequential(
    nn.Linear(input_dim, 512),
    nn.BatchNorm1d(512),
    nn.GELU(),
    nn.Linear(512, hidden_dim),
    nn.BatchNorm1d(hidden_dim),
    nn.GELU(),
    nn.Linear(hidden_dim, 2),
    nn.Linear(2, 1))
```

In our neural network configuration, the variable `hidden_dim` is empirically set to 32. The variable `input_dim`, however, varies depending on the specific ResNet architecture employed. Specifically, for ResNet-50, `input_dim` is set to 4098, whereas for both ResNet-34 and ResNet-18, `input_dim` is set to 1026.

## A.2 Evaluation

To evaluate the performance of the comparator $\mathbf{S}$ on the same task as during training, we employ the sampling procedure outlined in Fig. 4 on the test dataset.

For example, within the test set of **CUB-200**, which consists of 5,794 samples, we generate 110,086 pairs. Although we have already excluded the first nearest neighbor from the ground-truth class to avoid $x$ and $nn$ being similar, we still identify 740 pairs among these that consist of identical images, and then we remove them, resulting in a total of 109,346 pairs. This set includes 49,750 positive pairs and 59,596 negative pairs. In an effort to balance the distribution between positive and negative pairs, we eliminate an additional 9,846 negative pairs. This adjustment ensures that we are left with 99,500 pairs, achieving an exact 50/50 split between positive and negative pairs.

Following the same procedure on 8041 **Cars-196** test samples, we start with 152,779 pairs and remove 6,945 negative pairs, resulting in a balanced total of 145,834 pairs, with an exact 50/50 split between positives and negatives.

For **Dogs-120**, starting with 8,580 samples, we apply the same procedure to generate 163,020 pairs. To achieve balance, we eliminate 13,414 negative pairs. This results in a total of 149,606 pairs, ensuring a 50/50 split between positive and negative pairs.

The output score produced by the comparator $\mathbf{S}$ is then subjected to a threshold of 0.5 to classify each pair as either positive or negative.

## A.3 Architecture

Referring to Fig. 3, the written description of the architecture of the image comparator $\mathbf{S}$ are:

$$
\begin{aligned}
\mathbf{x}_{\text{conv1}} &= \mathbf{f}\left(\mathbf{x}\right); & \mathbf{x}_{\text{conv2}} &= \mathbf{f}\left(\mathbf{x}_{nn}\right) \\
\mathbf{x}_{\text{conv1}} &= \mathbf{flatten}\left(\mathbf{x}_{\text{conv1}}\right)^{T}; \mathbf{x}_{\text{conv2}} &= \mathbf{flatten}\left(\mathbf{x}_{\text{conv2}}\right)^{T} \\
\mathbf{x}_1 &= \left[\mathbf{x}_{cls}\|\mathbf{x}_{\text{conv1}}\right] + \mathbf{x}_{\text{pos}}; & \mathbf{x}_2 &= \left[\mathbf{x}_{cls}\|\mathbf{x}_{\text{conv2}}\right] + \mathbf{x}_{\text{pos}} \\
\mathbf{y}_1 &= \mathbf{x}_1 + \mathbf{MHSA}(\mathbf{x}_1); & \mathbf{y}_2 &= \mathbf{x}_2 + \mathbf{MHSA}(\mathbf{x}_2) \\
\mathbf{z1}, \mathbf{z2} &= \mathbf{CrossAttn}\left(\mathbf{y}_1, \mathbf{y}_2\right) \\
\mathbf{o} &= \mathbf{MLPs}\left(\mathbf{concat}\left(\mathbf{z}_1[0,:], \mathbf{z}_2[0,:]\right)\right) \\
\mathbf{s} &= \mathbf{sigmoid}\left(\mathbf{o}\right)
\end{aligned}
\tag{2}
$$

# B Hyperparameter tuning and ablation studies

In the following experiments, unless otherwise stated, we use iNaturalist-pretrained ResNet-50 Taesiri et al. (2022) for CUB-200 and ImageNet-pretrained ResNet-50 for Cars-196 and Dogs-120. Also, we evaluate design choices using binary classification accuracy (%) for the comparator $\mathbf{S}$ and top-1 classification accuracy for the re-ranking algorithm.

## B.1 At test time, comparing input with multiple nearest neighbors achieves better weights for re-ranking

**Considered datasets:** CUB-200 and Cars-196. **Considered tasks:** 200/196-way image classification.

In our re-ranking algorithm (Fig. 2), we only use one nearest neighbor (1 $nn$) to compare with the input image $x$ to compute the similarity score for a given class. In this section, we aim to study whether presenting

more nearest-neighbor (NN) samples to the comparator provides better class-wise re-ranking weights for the algorithm.

For each class, we perform the comparison between $x$ and $nn$ 2 or 3 times (i.e., pairing the input image $x$ with each of the 2 or 3 nearest neighbors from each class) and then take the average of these scores to determine the final weights. We then examine whether increasing the number of comparisons (by using more nearest neighbors) enhances the accuracy of the re-ranking process.

Table 5: Top-1 classification accuracy (%) using diffrent numbers of nearest neighbors. Using more nearest neighbors to compare with the input image slightly improves $\mathbf{C} \times \mathbf{S}$ .

| Number of nearest neighbors | CUB-200 Top-1 Acc (%) | Cars-196 Top-1 Acc (%) |
|---|---|---|
| 1 | 88.59 | 91.06 |
| 2 | 88.75 | 91.20 |
| 3 | 88.83 | 90.09 |

In Tab. 5, we find that increasing the number of nearest neighbors for comparison slightly improves the top-1 classification accuracy on both CUB-200 and Cars-196. With 2 nearest neighbors, the accuracy on the test set increases from 88.59% → 88.75% for CUB-200 and 91.06% → 91.20% for Cars-196. When using 3 nearest neighbors, the accuracy increases to 88.83% for CUB-200 and 90.09% for Cars-196. However, it is noteworthy to consider the associated computational cost, as it increases linearly with the number of nearest neighbors. As such, a careful balance between performance and computational efficiency should be struck.

### B.2 Using $Q = 10$ in sampling yields the optimal balance between the accuracy of S and the training cost

**Considered datasets:** CUB-200. **Considered tasks:** Binary classification.

$Q$ is the crucial hyperparameter for training $\mathbf{S}$ (see Sec. 3.4.3). To assess their effects beyond the default setting of $Q = 10$, we also train $\mathbf{S}$ with configurations of $Q = 3$, 5, and 15. This value represents the coverage of the sampling process over the nearest neighbor space.

Table 6: Binary classification accuracy (%) of $\mathbf{S}$ trained with different values of $Q$ for CUB-200. This value has a considerable influence on the final performance of $\mathbf{S}$. Using fewer than 10 significantly diminishes performance, while increasing to 15 offers only a marginal improvement at the expense of higher training costs. The training time complexity increases linearly with the value of $Q$, following the formula $2Q - 1$ (Fig. 4).

| Q | Train Binary Acc (%) | Test Binary Acc (%) |
|---|---|---|
| 3 | 97.17 | 87.49 |
| 5 | 97.60 | 92.94 |
| 10 | 98.69 | 94.44 |
| 15 | 99.30 | 94.62 |

However, large values of $Q$ could introduce noise during training. This is because NNs from the *tails* of the top-1 class (the last images in the middle row of Fig. 30a) and predicted classes (the last images in the bottom row of Fig. 30a) significantly diverge from the *head* samples used in comparison with the query image during test time.

From Tab. 6, we notice that these values significantly influence the performance of $\mathbf{S}$. Interestingly, there appears to be a trend: as these values increase (indicating more training data), the test accuracy also goes up (seemingly saturated after 10). However, similar to Sec. B.1, there is a balance to be considered between test accuracy and the computational cost.

### B.3 Using $K = 10$ in re-ranking yields the optimal classification accuracy

**Considered datasets:** CUB-200 and Dogs-120. **Considered tasks:** 200/120-way image classification.

Given the optimal value of $Q$ being 10 for training image comparators **S** demonstrated in Appendix B.2, we tested various $K$ values for the re-ranking algorithm described in Sec. 3.3. Empirically, we found that $K = 10$ delivers the best classification accuracy on both CUB-200 and Dogs-120 datasets. Hence, we use $K = 10$ throughout our main experiments in the main text.

Table 7: Performance and runtime comparison for different values of $K$ on CUB-200 and Dogs-120 datasets.

| K | CUB-200 Top-1 Acc (%) | Runtime (s) | Dogs-120 Top-1 Acc (%) | Runtime (s) |
|---|---|---|---|---|
| 1 | 85.83 | 8.81 | 85.82 | 8.81 |
| 2 | 87.95 (+2.12) | 27.72 | 86.06 (+0.24) | 50.35 |
| 3 | 88.28 (+2.45) | 32.32 | 86.03 (+0.21) | 54.96 |
| 5 | 88.28 (+2.45) | 41.53 | 85.91 (+0.09) | 64.16 |
| 10 | 88.42 (**+2.59**) | 64.55 | 86.27 (**+0.45**) | 87.18 |
| 15 | 88.00 (+2.17) | 87.57 | 85.86 (+0.04) | 110.20 |

### B.4 Using C's predicted labels to sample hard negative pairs is more beneficial than using random negative pairs

**Considered datasets:** CUB-200. **Considered tasks:** Binary classification & 200-way image classification.

We run two experiments. **First**, we replace our proposed negative samples (from top-2 to top-$Q$ class; see Fig. 2a) by random samples.

In our data sampling algorithm (Fig. 4), the positive and negative classes are determined by a classifier **C** and it is intriguing to study whether we can avoid the reliance on **C** to choose the classes for nearest neighbor retrievals. Therefore, we modify the data sampling algorithm in Sec. 3.4 so that for a training image $x$, **the NNs for positives are from the ground-truth class and the NNs for negatives are randomly picked from the remaining class**. We train an image comparator for CUB-200 with the same settings used for the one in Tab. 2.

In this setup, negative pairs become much easier to classify due to their stark visual dissimilarity, leading **S** to predominantly learn from positive pairs. Indeed, **S** demonstrates limited ability to identify the misclassifications made by the ResNet-50. When this image comparator **S** is utilized in the re-ranking algorithm (Fig. 2) for 200-way image classification, there is a notable decline in the top-1 accuracy from 88.59% → 86.55%. This result confirms the novelty of our sampling process and proves that hard negatives (close-species pairs in Sec. 3.4) are the key factor to help **S** achieve competitive performance. Please note that in this experiment, the input sample is still compared with its first nearest neighbors in test.

**Second**, we use the 2nd or 3rd closest nearest neighbors (NNs) in each class instead of using the 1st to train **S**. We find that using the 2nd and 3rd NNs in the sampling process (both in training and test) is suboptimal, decreasing the top-1 accuracy from 88.59% → 88.06% and 88.21%, respectively.

**B.5  Simple and parametric similarity functions are insufficient to distinguish fine-grained species**

**Considered datasets:**  CUB-200. **Considered tasks:**  200-way image classification.

To understand the significance of image comparator **S** for re-ranking, we consider three alternatives in Tab. 8: (a) cosine similarity – non-parametric, image-level, (b) Earth Mover's Distance (EMD): non-parametric, patch-level, and (c) 4-layer MLP (Appendix A): parametric. We use avgpool features for cosine and layer4's conv features for EMD and the 4-layer MLP. All these three functions fully manage the re-ranking, akin to the role of **S** in **C → S**.

(a) We replace **S** with cosine similarity function, which chooses the class that yields the highest similarity to the input, among top-$Q$ categories ranked by ResNet-50, as the final prediction. However, we find this degrades the top-1 accuracy from 87.72% → 60.20%. (b) We use EMD function from Zhang et al. (2020) to also pick the closest class to the input but observe a drop from 87.72% → 54.83%. (c) We remove Transformer layers in Fig. 3 and train a comparator network (with **f** being frozen) following the same settings used for **S** in Sec. 3. Yet, this simple comparator leads to a reduced accuracy of 83.76%.

Table 8: Top-1 classification accuracy on CUB-200 when we use 2nd or 3rd NN samples for negative pairs instead of the 1st NNs as depicted in Fig. 4. For our re-ranking algorithm, 1st is optimal. We also experiment with different similarity functions for re-ranking and find that comparator **S** is the best.

| NN-th | RN50 × **S** | **C → S**  re-ranking on CUB-200 (%) | | | |
|---|---|---|---|---|---|
| | | Our trained **S** | cosine | EMD | 4-layer MLP |
| 1st | 88.59 | 87.72 | 60.20 | 54.83 | 83.76 |
| 2nd | 88.06 | 87.34 | 58.84 | 57.05 | 84.33 |
| 3rd | 88.21 | 87.43 | 57.47 | 57.14 | 83.93 |

Below we show examples where cosine cannot assign distinctively high similarity scores for groundtruth classes among other top-probable classes, thus being unable to re-rank RN50 predictions effectively.

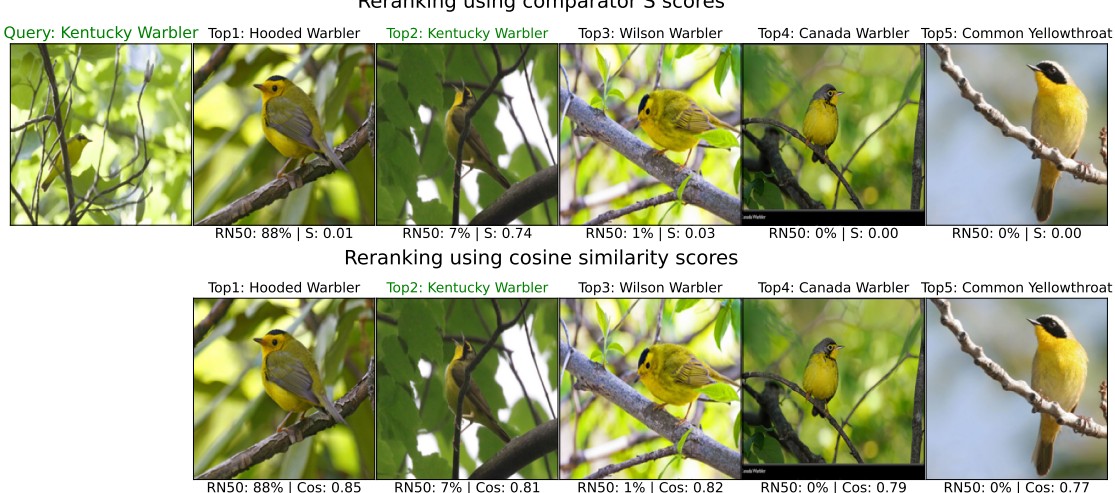

Figure 9: A pretrained RN50 model predicts the Query image's class (ground-truth label: American Crow) and produces an initial ranking, but the top-1 predicted class (Shiny Cowbird: 88%) does not match the ground-truth. Re-ranking using scores from the comparator **S** alone correctly identifies the top-1 class as American Crow, whereas cosine similarity scores fail to do so.

Figure 10: A pretrained RN50 model predicts the Query image's class (ground-truth label: Kentucky Warbler) and produces an initial ranking, but the top-1 predicted class (Hooded Warbler: 88%) does not match the ground-truth. Re-ranking using scores from the comparator **S** alone correctly identifies the top-1 class as Kentucky Warbler, whereas cosine similarity scores fail to do so.

### B.6 Finetuning pretrained conv features is important in training S

**Considered datasets:** CUB-200. **Considered tasks:** Binary classification.

In the training process of **S**, one crucial consideration is whether to optimize the pretrained conv layers for the new task. If these layers are already generating effective encodings for the binary classification task, then we should focus on optimizing the remaining layers. To evaluate the effects of retraining the convolutional (conv) layers, we train an image comparator **S** for CUB-200, ensuring that the conv layers remained frozen while all other settings were maintained as they were specified in Sec. A.

We find that freezing the conv layers significantly slows down the learning process of **S**. Specifically, the model could not converge after 100 epochs (training accuracy of 94.18%) and the test accuracy of 91.94% (lower than 94.44% when making conv layers trainable in Tab. 5).

### B.7 The performance of S and re-ranking algorithm is statistically consistent

**Considered datasets:** CUB-200, Cars-196, and Dogs-120. **Considered tasks:** Binary classification & 200/196/120-way image classification.

To ensure the statistical significance of **S**'s performance, we train it multiple times on CUB-200 and Cars-196 using various random seeds and present the results in Tab. 9.

Table 9: Binary classification accuracy of **S** (%) and top-1 classification accuracy (%) of Product of Experts **C** × **S** over 3 random seeds. The performance of **S** and our re-raking algorithm **C** × **S** is statistically consistent.

| # | Train Binary Acc (%) | | | Test Binary Acc (%) | | | **C** × **S** Top-1 Acc (%) | | |
|---|---------|----------|----------|---------|----------|----------|---------|----------|----------|
|   | CUB-200 | Cars-196 | Dogs-120 | CUB-200 | Cars-196 | Dogs-120 | CUB-200 | Cars-196 | Dogs-120 |
| 1 | 98.69 | 99.01 | 98.83 | 94.37 | 94.97 | 92.01 | 88.42 | 90.86 | 86.27 |
| 2 | 98.66 | 98.99 | 98.79 | 94.53 | 95.07 | 92.09 | 88.82 | 91.23 | 86.33 |
| 3 | 98.61 | 99.02 | 98.81 | 94.44 | 95.04 | 92.96 | 88.52 | 91.08 | 86.32 |
|   | 98.65±0.04 | 99.00±0.02 | 98.81±0.02 | 94.44±0.06 | 95.02±0.04 | 92.05±0.03 | 88.59±0.17 | 91.06±0.15 | 86.31±0.02 |

We see that **S** consistently achieves similar test accuracies for both binary and single-label classification tasks across different runs. Further, we note that **S** misclassifies pairs on CUB-200 and Cars-196. We later find in Sec. I.2 that these errors are primarily due to inter-class and intra-class variations, as well as issues with dataset annotations.

### B.8 Removing data augmentation degrades the performance of S

**Considered datasets:** CUB-200, Cars-196, and Dogs-120. **Considered tasks:** Binary classification.

We wish to study the impact of data augmentation on the performance of **S** by repeating our previous experiments for CUB-200 and Cars-196 datasets without applying TrivialAugment (Müller & Hutter, 2021). In Tab. 10, we observe that the model not only converges more quickly in **training** without augmentation but also attains higher accuracies over the training set.

Table 10: Binary classification accuracy (%) of **S** on CUB-200 and Cars-196 with and without Trivial Augmentation Müller & Hutter (2021) in training. Data augmentation from Trivial Augmentation plays an important role in the performance of **S**.

| Phase | CUB-200 | | Cars-196 | | Dogs-120 | |
|-------|------|---------|------|---------|------|---------|
|       | with | without | with | without | with | without |
| Train Binary Acc (%) | 98.65 | 99.81 | 99.00 | 99.79 | 98.81 | 99.82 |
| Test Binary Acc(%) | 94.44 | 93.12 | 95.02 | 93.50 | 92.05 | 90.84 |

Nonetheless, the omission of data augmentation has a significantly negative effect during test time (i.e., hurting model robustness). **S** achieves 94.44% with data augmentation, while the omission results in a lower accuracy of 93.12% for CUB-200. Similarly on Cars-196 and Dogs-120, we observe drops in test accuracy without using the data augmentation technique, from 95.02% → 93.50% and 92.05% → 90.84%. These findings underscore the importance of data augmentation in the training of the image comparator, as it helps combat overfitting. This finding comes as no surprise, as the effectiveness of data augmentation in in-the-wild datasets has been well-proven in literature Nguyen et al. (2023).

### B.9 The importance of the finetuned feature f, self-attention, cross-attention, and MLP in comparator S

**Considered datasets:** CUB-200. **Considered tasks:** 200-way image classification.

To study the effects of different image comparators **S** on re-ranking, we unfolds a systematic exploration of architectural configurations, revealing that our proposed architecture in Fig. 3 is optimal. We attempt to re-rank the top-$Q$ CUB-200 categories given by an iNaturalist-pretrained ResNet-50.

We begin with a simple baseline, where the comparator **S** in **C** → **S** is substituted by a cosine similarity function. However, this approach only yields an accuracy of 60.20%.

Next, while keeping the pretrained conv layers **f** still frozen, we train a comparator **S** consisting of the frozen **f** followed by a 4-layer MLP defined in Sec. A.1 to distinguish positive and negative pairs – the task described in Sec. 3.4.2. Using **C** → **S** for re-ranking, we see an increase from 60.20% → 83.76%. Despite this improvement, the achieved accuracy does not surpass that of the standalone pretrained classifier **C**, which has an accuracy of 85.83%.

We then proceed to fine-tune the pretrained convolutional layers **f** in tandem with the 4-layer MLP. This fine-tuning process enables **f** to better adapt its learned representations for the image comparison task. As a result, we observe an enhancement in accuracy from 83.76% → 87.31%, marking an improvement over the pretrained ResNet-50 by +1.48 pp.

Finally, we integrate transformer layers into our model as depicted in Fig. 3, and we train **S** using training settings that are consistent with those previously described. This helps **S** further refine its performance, elevating the accuracy from 87.31% → 87.72.

## C Well-trained comparator S works well with an arbitrary classifier C

**Considered datasets:** CUB-200. **Considered tasks:** 200-way image classification.

We have shown that using a comparator **S** trained with a classifier **C** improves the classifier's performance. Here we wish to investigate whether a well-trained compartor **S** can be used with other arbitrary black-box, pretrained image classifiers to improve them through our re-ranking algorithms. We refer to these classifiers as "unseen models" to distinguish them from the classifier **C** that **S** has worked with during training.

Table 11: Top-1 classification accuracy (%) of classifiers seen and unseen during the training of **S**. When coupling black-box, unseen classifiers with **S**, we see significant and consistent accuracy boosts on CUB-200.

| Model | Seen | Unseen | | | | |
| | iNat-RN50 | IN1K-RN18 | IN1K-RN34 | IN1K-RN50 | ViT-B-16 | NTS-Net |
|---|---|---|---|---|---|---|
| **C** | 85.83 | 60.22 | 62.81 | 62.98 | 82.40 | 87.04 |
| **C** → **S** | 87.73 | 84.38 | 84.74 | 84.52 | 87.66 | 87.94 |
| **C** × **S** | 88.42 | 83.60 | 83.83 | 83.62 | 88.45 | 87.14 |

In Tab. 11, we denote **C** as the pretrained classifiers, while **C** → **S** and **C** × **S** represent our proposed re-ranking algorithms, detailed further in Section D. The notation IN1K refers to models pretrained on the ImageNet-1K dataset Deng et al. (2009). Using an image comparator **S**, well-trained with the iNaturalist-pretrained ResNet-50 (iNat-RN50), we observe significant improvements in the accuracy of classifiers not

25

seen during training. Specifically, improvements are up to 23.28 pp for $\mathbf{C} \times \mathbf{S}$ and 24.16 pp for $\mathbf{C} \rightarrow \mathbf{S}$ when applied to IN1K-RN18.

# D  Re-ranking algorithms for single-label image classification

## D.1  Baseline classifiers

**ResNet:** In Sec. 3.3, we introduced re-ranking algorithms that reorder the top-predicted classes of a pretrained classifier, denoted as **C**. Given a pretrained ResNet, we wish to investigate whether our classifier can improve over the classifier **C** working alone.

**k-Nearest Neighbors (kNN):** kNN classifiers operate by comparing an input image against the whole training dataset at the image level (image-to-image comparison). Our classifier considers only a small, selective subset of the training samples for this comparison.

## D.2  Baseline: Re-ranking using similarity scores of S alone

In Algorithm 1 (see illustration in Fig. 11), we directly leverage the similarity scores of **S** to rank $Q$ class labels. Specifically, upon receiving an image $x$, **C** assigns all $k$ possible classes with softmax scores. The set of top-ranked classes **T** and their respective scores is then selected for re-ranking. For each class, **S** compares $x$ with the representative image (i.e., nearest neighbor) of that class and generates a similarity score (from $0.0 - 1.0$), indicating the likelihood that the two images belong to the same category. The algorithm then reorders **T** based on the scores generated by **S** and gives the new top-1 predicted label $\hat{y}$.

---

**Algorithm 1 Hard** re-ranking using the ranking of the similarity scores returned by **S** alone

**Inputs**: Image $x$, Pretrained classifier **C**, Image comparator **S**, Number of top classes to consider $Q$.
**Output**: New ranking of classes **T**′ and top-1 label $\hat{y}$.

1:  // Get initial predictions and scores from **C**
2:  $\mathbf{P}, \mathbf{scores} \leftarrow \mathbf{C}(x)$
3:  // Select top-$Q$ classes and scores
4:  $\mathbf{T}, \mathbf{scores}_{\text{top}} \leftarrow \text{SelectTopQ}(\mathbf{P}, \mathbf{scores}, Q)$
5:  // List to store re-ranked classes with their final scores
6:  $\mathbf{R} \leftarrow$ empty list
7:  **for** $i = 1$ to $Q$ **do**
8:      // Obtain nearest-neighbor explanation for the class
9:      $\text{NN\_image} \leftarrow \text{RetrieveNearestNeighbor}(x, \mathbf{T}[i])$
10:     // Compute similarity score of **S** on the class
11:     $s_i \leftarrow \mathbf{S}(x, \text{NN\_image})$
12:     // Update the final score for the class
13:     $s_{\text{final}} \leftarrow s_i$
14:     // Store the class and its respective final score
15:     $\mathbf{R}.\text{append}(\mathbf{T}[i], s_{\text{final}})$
16:  **end for**
17:  // Re-rank the classes based on final scores
18:  $\mathbf{T}' \leftarrow$ Sort $\mathbf{R}$ by final scores in descending order
19:  // Get the new top-1 predicted class for **C**
20:  $\hat{y} \leftarrow$ First element of $\mathbf{T}'$
21:
22:  **return**  $\mathbf{T}'$ and $\hat{y}$

---

## D.3  Soft re-ranking using product between scores of C and S

In Algorithm 2, we combine the scores from the pretrained classifier **C** and the comparator network **S**. For each image $x$, after **C** assigns initial scores to the top $Q$ classes, these scores are then recalculated by multiplying with the confidence scores generated by **S**. This algorithm represents a more nuanced approach, leveraging the strengths of both **C** and **S** to improve the classification accuracy.

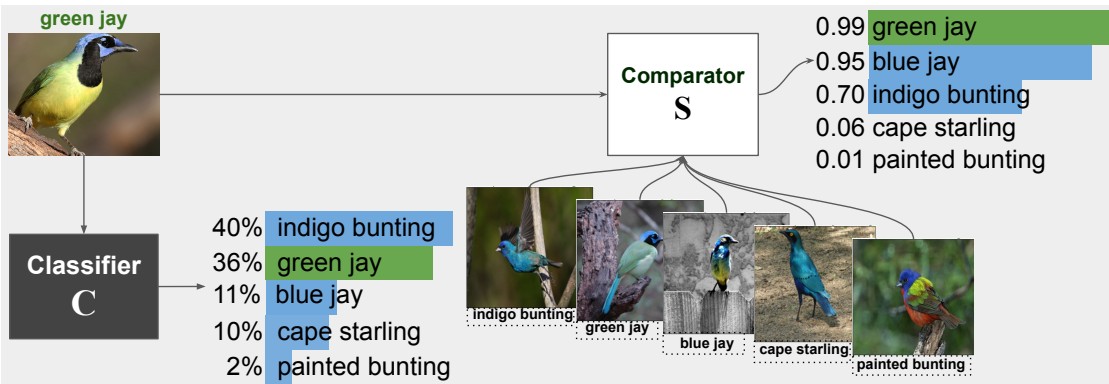

Figure 11: **Hard re-ranking.** Re-ranking top-predicted classes of a pretrained classifier $\mathbf{C}$ using the ranking of the similarity scores returned by $\mathbf{S}$ ($\mathbf{C} \rightarrow \mathbf{S}$). No Product of Experts.

---

**Algorithm 2 Soft** Re-ranking using Product of Experts ($\mathbf{C} \times \mathbf{S}$ )

---

**Inputs**: Image $x$, Pretrained classifier $\mathbf{C}$, Image comparator $\mathbf{S}$, Number of top classes to consider $Q$.
**Output**: New ranking of classes $\mathbf{T'}$ and top-1 label $\hat{y}$.

1: // Get initial predictions and scores from $\mathbf{C}$
2: $\mathbf{P}, \mathbf{scores} \leftarrow \mathbf{C}(x)$
3: // Select top-$Q$ classes and scores
4: $\mathbf{T}, \mathbf{scores}_{\text{top}} \leftarrow \text{SelectTopQ}(\mathbf{P}, \mathbf{scores}, Q)$
5: // List to store re-ranked classes with their final scores
6: $\mathbf{R} \leftarrow$ empty list
7: // Loop over top-$Q$ classes in $\mathbf{T}$ and their scores
8: **for** $i = 1$ to $Q$ **do**
9:     // Obtain nearest-neighbor explanation for the class
10:     NN_image $\leftarrow \text{RetrieveNearestNeighbor}(x, \mathbf{T}[i])$
11:     // Compute similarity score of $\mathbf{S}$ on the class
12:     $s_i \leftarrow \mathbf{S}(x, \text{NN\_image})$
13:     // Compute final score as the product of initial class score and similarity score of $\mathbf{S}$
14:     $s_{\text{final}} \leftarrow \mathbf{scores}_{\text{top}}[i] \times s_i$
15:     // Store the class and its respective final score
16:     $\mathbf{R}.\text{append}(\mathbf{T}[i], s_{\text{final}})$
17: **end for**
18: // Re-rank the classes based on final scores
19: $\mathbf{T'} \leftarrow$ Sort $\mathbf{R}$ by final scores in descending order
20: // Get the new top-1 predicted class for $\mathbf{C}$
21: $\hat{y} \leftarrow$ First element of $\mathbf{T'}$
22:
23: **return** $\mathbf{T'}$ and $\hat{y}$

---

We visually illustrate the algorithm in the Fig. 2.

# E    Computational costs of image classifiers

Although our proposed classifier yields significant improvements in top-1 classification accuracy on tested datasets, it also introduces additional computational overhead beyond that of the pretrained classifier $\mathbf{C}$. To quantify the computational expense of our product-based classifier in Tab. 1, we run each model (listed in Tab. 12) 5 times, processing 1,000 queries on a single NVIDIA V100 GPU.

Referring to Algorithm 2, the total runtime of our classifier can be represented as:

$$T = T_{RN50} + T_{kNN} + T_{\mathbf{S}}. \tag{3}$$

Here, $T_{RN50}$ denotes the time taken by the classifier $\mathbf{C}$ (i.e., ResNet-50), $T_{kNN}$ represents the time required for nearest-neighbor retrieval, and $T_{\mathbf{S}}$ is the time taken by the image comparator $\mathbf{S}$ to compare the query image with all top-$Q$ classes.

In Tab. 12, we find that our classifier is marginally slower than RN50, kNN, ProtoPNet, and Deformable-ProtoPNet. However, it surpasses these models in terms of top-1 accuracy. Additionally, we note that our classifier is significantly faster compared to correspondence-based classifiers such as EMD-Corr and CHM-Corr, and it also demonstrates higher accuracy.

Table 12: Average runtime (s) of classifiers. Our re-ranking algorithm runs slower compared to $\mathbf{C}$, kNN, and Proto-part classifiers, yet significantly faster than visual-corr classifiers.

| Model | Runtime (s) |
|---|---|
| RN50 | $8.81 \pm 0.14$ |
| kNN | $9.70 \pm 0.32$ |
| EMD-Corr | $1927.69 \pm 17.48$ |
| CHM-Corr | $6920.76 \pm 67.58$ |
| ProtoPNet | $9.78 \pm 0.20$ |
| Deformable-ProtoPNet | $9.98 \pm 0.26$ |
| $\mathbf{S}$ | $46.04 \pm 0.04$ |
| RN50 $\times\mathbf{S}$ | $64.55 \pm 0.35$ |

## E.1    Improving runtime of $\mathbf{C} \times \mathbf{S}$  with reduced number of queries to image comparator S

To mitigate the additional computational overhead introduced by image comparator $\mathbf{S}$, we aim to reduce the number of queries to $\mathbf{S}$ by ignoring less probable labels. In Algorithm 2, for each input image, we set the queries to K = 10, always examining the **top-10** most probable classes. This is sub-optimal for model efficiency. Specifically, there are always classes receiving less than 1% probability by ResNet-50, which are unlikely to be the top-1 after re-ranking(see Fig. 2). Reducing the number of $K$ can significantly save computation with possibly minimal impact on accuracy.

To answer this, we conduct an experiment on CUB-200 where, instead of re-ranking the entire **top-10**, we only re-rank the classes that have a probability of 1% or higher assigned by the base classifier $\mathbf{C}$. We call this method **thresholding**.

We found that:

- $\mathbf{C} \times \mathbf{S}$  model performance on CUB-200 dropped very marginally by **only 0.08%** (from 88.59% to 88.51%).

- The number of queries to image comparator $\mathbf{S}$ was reduced by approximately **4x** (from 10 to about 2.5 queries per image).

This results in a **2.5x** speedup in the overall runtime of the $\mathbf{C} \times \mathbf{S}$  model (from 64.55 seconds to 28.95 seconds), as shown in the Tab. 13.

Table 13: Comparison of model performance and runtime with and without thresholding for re-ranking.

| Model | Runtime (s) | Top-1 Acc (%) |
|---|---|---|
| RN50 xS | 64.55 ± 0.35 | 88.59 |
| **RN50 xS (with threshold)** | 28.95 ± 0.11 | 88.51 |

### E.2 Improving runtime of C × S with reduced training set during inference

The need for the entire training dataset at test time for nearest neighbor retrieval is a limitation of our work. In this section, we aim to study the impact of reducing the size of the training data on inference-time performance. In particular, we shrink the training set from 100% to 50% and 33% of the original size and record the effect on the accuracy and runtime. The samples being removed are randomly selected from the training set.

Table 14: The top-1 accuracy and runtime of **C × S** on CUB-200 and Dogs-120 for different sizes of training data during inference.

| Dataset | % Data | Samples per Class | Top-1 Acc (%) | Runtime (s) |
|---|---|---|---|---|
| CUB-200 | 100 | 30 | 88.43 | 64.55 |
| CUB-200 | 50 | 15 | 88.26 | 59.70 |
| CUB-200 | 33 | 10 | 88.19 | 58.08 |
| Dogs-120 | 100 | 100 | 86.27 | 87.18 |
| Dogs-120 | 50 | 50 | 86.32 | 71.02 |
| Dogs-120 | 33 | 33 | 86.42 | 65.52 |

We found in Tab. 14 that reducing the size of the training data has little-to-no impact on the inference-time performance. When keeping the same accuracy, we can reduce the runtime by 10% on CUB-200 and 24.9% on Dogs-120 by reducing the training set to 33% of the original size. It is an interesting research question to determine the smallest training set size that still maintains the same accuracy. However, as this question is orthogonal to the main focus of our work, we opt to leave it for future work.

## F Sanity checks for image comparator S

### F.1 Controlling nearest-neighbor explanations

**Considered datasets:** CUB-200, Cars-196, and Dogs-120. **Considered tasks:** Binary image classification.

In this section, we investigate the logical consistency of **S**. Specifically, we evaluate the model's ability to correctly distinguish images that are the same and different. This is achieved by comparing **S**'s responses when the NN is the input query itself (expected output: 1 or **Yes**) and when the NN is a randomly sampled image (expected output: 0 or **No**). When sampling the random NN, we study two scenarios: one where NNs are assigned random values and another where they are random *real* images. These random *real* images are sampled by shuffling the NNs in image pairs inside a batch size of 64 pairs. For image comparators **S**, we use iNaturalist-pretrained ResNet-50 for CUB-200 and ImageNet-pretrained ResNet-50 for both Cars-196 and Dogs-120.

As seen in Tab. 15, when the NN provided is the input image, **S** correctly recognizes the two images as being from the same class, yielding a **Yes** ratio of 100%. On the other hand, when a random-valued NN is used, the comparator network correctly identifies that the images are not from the same class, yielding **low Yes** ratios, namely 3.51%, 2.01%, and 1.86% for CUB-200, Cars-196, and Dogs-120 , respectively. The ratios represented by **Yes** for random-real images are also low for both datasets, as expected. This verifies the logical consistency of **S**.

Table 15: Logical tests for the image comparator **S** by controlling nearest-neighbor explanations. These tests involve measuring the ratio (%) at which **S** perceives $x$ and its nearest neighbor $nn$ as belonging to the same class. **S** performs as expected in these sanity checks. Images with random values, being out-of-distribution samples for **S**, sometimes lead to incorrect outputs.

| Explanation | CUB-200 (%) | Cars-196 (%) | Dogs-120 (%) |
|---|---|---|---|
| Query | **100** | **100** | **100** |
| Random values | **3.51** | **2.01** | **7.93** |
| Random real images | **0.71** | **0.79** | **1.86** |

### F.2 Comparator S tends to be more confidence when it is correct than when it is wrong

**Considered datasets:** CUB-200. **Considered tasks:** Binary image classification.

In analyzing the performance of image comparator **S** on CUB-200, we observe intriguing insights into its similarity scores. In particular, we are interested in 4 types of classifications made by **S** in the binary (Accept/Reject) task: Correctly Accepting, Incorrectly Accepting, Correctly Rejecting, and Incorrectly Rejecting the top-1 predicted label of an iNaturalist-pretrained ResNet-50 classifier. The average confidence scores for these cases are as follows: "CorrectlyAccept" at 99.08%, "IncorrectlyAccept" at 91.85%, "CorrectlyReject" at 93.50%, and "IncorrectlyReject" at 87.09%.

These numbers reveal a nuanced understanding of the image-comparator network's inner-workings. More specifically, **S** demonstrates higher scores in its correct decisions (either Accept or Reject) compared to the incorrect ones. This characteristic makes **S** compatible for re-weighting initial predictions of a classifier **C** using **S** scores. When the comparator accepts a label, it is more likely to be groundtruth, and conversely for rejections, thus allowing effective up-weighting for correct labels and down-weighting for incorrect labels.

This observation inspires us to update confidence scores initially given by pretrained classifier **C** using product of model scores. Formally, for any given class **A**, the pretrained classifier **C** assigns a confidence score, denoted as $s_{C_A}$, and the comparator network **S** assigns score $s_{S_A}$. The final score $s_A$ for the class is updated to:

$$s_A = s_{C_A} \times s_{S_A} \tag{4}$$

We incorporate this product of scores in Algorithm 2.

## G  Maximum possible accuracy with re-ranking algorithms

In this section, we present the maximum achievable accuracy of classifiers **C** in Tab. 2, 3, and 4 to explore the potential of re-ranking algorithms like the one illustrated in Fig.2. Surprisingly, by merely considering the top-1 and top-2 probable classes, we consistently observe accuracy enhancements ranging from $6 \rightarrow 9$ pp in Tab. 16. As we extend our consideration to higher-ranked categories, the gains in accuracy plateau, ultimately approaching near-perfect accuracy at the top 10.

Table 16: Maximum achievable classification accuracy (%) with RN50 classifiers in Tab. 2, 3, and 4. Re-ranking algorithms can achieve up to approx. 99% when considering the top-10 predicted labels with highest scores. The results indicate significant potential for improving classification accuracy via re-ranking top-predicted classes like Fig. 2.

| Top-$Q$ | CUB-200 | Cars-196 | Dogs-120 |
|---|---|---|---|
| 1 | 85.86 | 89.73 | 85.84 |
| 2 | 93.10 | 96.18 | 94.59 |
| 3 | 95.53 | 97.77 | 97.10 |
| 4 | 96.84 | 98.48 | 98.09 |
| 5 | 97.48 | 98.79 | 98.74 |
| 6 | 97.84 | 98.99 | 99.04 |
| 7 | 98.10 | 99.12 | 99.22 |
| 8 | 98.34 | 99.19 | 99.30 |
| 9 | 98.50 | 99.29 | 99.36 |
| 10 | 98.64 | 99.38 | 99.49 |

## H   Evaluating probable-class nearest-neighbor explanations on humans

We investigate whether PCNN explanations, learned by the image comparator $\mathbf{S}$, offer more assistance to humans than traditional top-1 nearest neighbors Nguyen et al. (2021); Taesiri et al. (2022) in the binary decision-making task. For simplicity, we refer to these as the top-1 and top-$Q$ experiments.

**Query samples**   For each of the top-1 and top-$Q$ experiments, we select 300 correctly classified and 300 misclassified query samples determined by CUB-200 RN50 × $\mathbf{S}$, amounting to a total of 600 images. From this pool, a human user is presented with a randomly chosen subset of 30 images.

**Visual explanations**   In the top-$Q$ experiment, we retrieve one nearest-neighbor prototype from each of the multiple top-ranked classes as determined by $\mathbf{C} \times \mathbf{S}$ and then present these to human participants (see Fig. 14). By contrast, in the top-1 experiment, human users are only shown the nearest neighbors from the single top-1 predicted class (see Fig. 13). In both experiments, participants are presented with the input image alongside 5 nearest neighbors. Their task remains the same: to accept or reject the top-1 predicted label from $\mathbf{C} \times \mathbf{S}$, utilizing the explanations provided. To minimize potential biases, we do not display the confidence scores for the classes.

**Participants**   There are 35 voluntary participants in the top-1 experiment and 25 in the top-$Q$ experiment, respectively. Although we do not implement formal quality control measures, such as filtering out malicious users, we encourage participants to perform responsibly. Moreover, we consider only the data from participants who complete all 30 test trials.

**Study interface**   Prior to the experiments, participants are briefed on the task with instructions as detailed in Fig. 12. Each user is tasked with 30 binary questions for an experiment. You can also try out yourself the human experiment interface at this **link**.

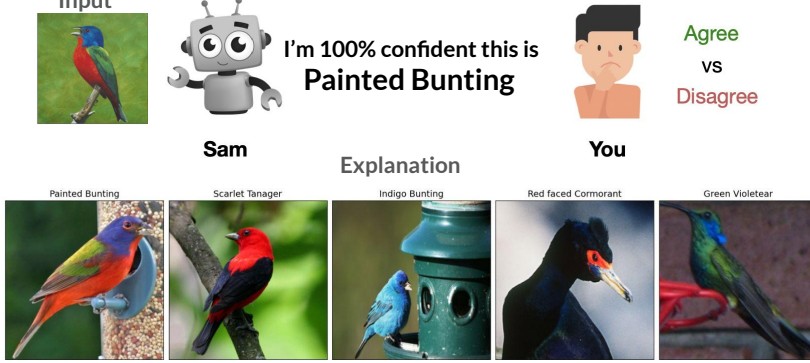

Figure 12: The experiment instructions shown to voluntary participants.

**Accept/Reject AI predicted label using Explanations**

| Username | Labeled Images | Total Images | Type of NNs |
|---|---|---|---|
| me-u2h1saiu | 1 | 30 | top1 |

Sam guessed the Input image is **Painted Bunting** with **10%** confidence.
Is this bird a **Painted Bunting**?

| YES | NO |
|---|---|

**Accept/Reject AI predicted label using Explanations**

| Username | Labeled Images | Total Images | Type of NNs |
|---|---|---|---|
| me-u2h1saiu | 9 | 30 | top1 |

Sam guessed the Input image is **Tropical Kingbird** with **100%** confidence.
Is this bird a **Tropical Kingbird**?

| YES | NO |
|---|---|

**Accept/Reject AI predicted label using Explanations**

| Username | Labeled Images | Total Images | Type of NNs |
|---|---|---|---|
| me-u2h1saiu | 14 | 30 | top1 |

Sam guessed the Input image is **Myrtle Warbler** with **90%** confidence.
Is this bird a **Myrtle Warbler**?

| YES | NO |
|---|---|

Figure 13: Example test trials given to human users for the top-1 experiment. Nearest neighbors are retrieved from the top-1 predicted class of a CUB-200 ResNet-50 classifier.

**Accept/Reject AI predicted label using Explanations**

| Username | Labeled Images | Total Images | Type of NNs |
|---|---|---|---|
| username-2e3k8iz3 | 10 | 30 | topQ |

Sam guessed the Input image is **Brewer Blackbird** with **17%** confidence.
Is this bird a **Brewer Blackbird**?

| YES | NO |
|---|---|

**Accept/Reject AI predicted label using Explanations**

| Username | Labeled Images | Total Images | Type of NNs |
|---|---|---|---|
| username-73z66pn2 | 2 | 30 | topQ |

Sam guessed the Input image is **Indigo Bunting** with **100%** confidence.
Is this bird a **Indigo Bunting**?

| YES | NO |
|---|---|

**Accept/Reject AI predicted label using Explanations**

| Username | Labeled Images | Total Images | Type of NNs |
|---|---|---|---|
| username-73z66pn2 | 12 | 30 | topQ |

Sam guessed the Input image is **Purple Finch** with **99%** confidence.
Is this bird a **Purple Finch**?

| YES | NO |
|---|---|

Figure 14: Example test trials given to human users for the top-$Q$ experiment. Nearest neighbors are retrieved from the multiple top-ranked classes of a CUB-200 ResNet-50 classifier.

**Main findings**  In Tab. 17, we find that showing human participants with probable-class nearest-neighbor explanations enhances their accuracy by around +10 pp, boosting it from 54.55% (± 9.54) to 64.58% (± 8.06), in the task of distinguishing correct from incorrect classifications made by the classifier $\mathbf{C} \times \mathbf{S}$ . Additionally, we observe that participants with top-1 explanations tend to accept predictions at a very high rate of up to 85.15%, inline with prior studies indicating that visual explanations can increase human trust, yet may also result in higher false positive rates Nguyen et al. (2021); Fok & Weld (2023).

In contrast, those with top-$Q$ explanations accept predictions at a lower 64.81%. This trend in acceptance rates correlates with the performance patterns in Fig. 8, where top-1 participants exhibit high accuracy with correct AI decisions and lower accuracy with incorrect ones.

Table 17: Human accuracy (%) upon AI correctness on test samples for top-1 and PCNN and data statistics.

| Explanation | AI Correctness | $\mu$ (%) | $\sigma$ (%) | Numb. of Samples |
|---|---|---|---|---|
| top-1 | AI is Wrong | 22.28 | 15.02 | 525 |
| top-1 | AI is Correct | 90.99 | 7.73 | 465 |
| Overall | — | 54.55 | 9.54 | 990 |
| PCNN | AI is Wrong | 49.31 | 17.27 | 404 |
| PCNN | AI is Correct | 79.78 | 14.46 | 406 |
| Overall | — | 64.58 | 8.06 | 810 |

# I Visualizations

## I.1 Visualization of top-1 corrections via re-ranking in single-label image classification

We present cases where $\mathbf{C} \times \mathbf{S}$ corrects the top-1 predictions made by pretrained classifiers $\mathbf{C}$ on CUB-200 (Figs. 15 – 18), Cars-196 (Figs. 19 –22), Dogs-120 (Figs. 23 –26).

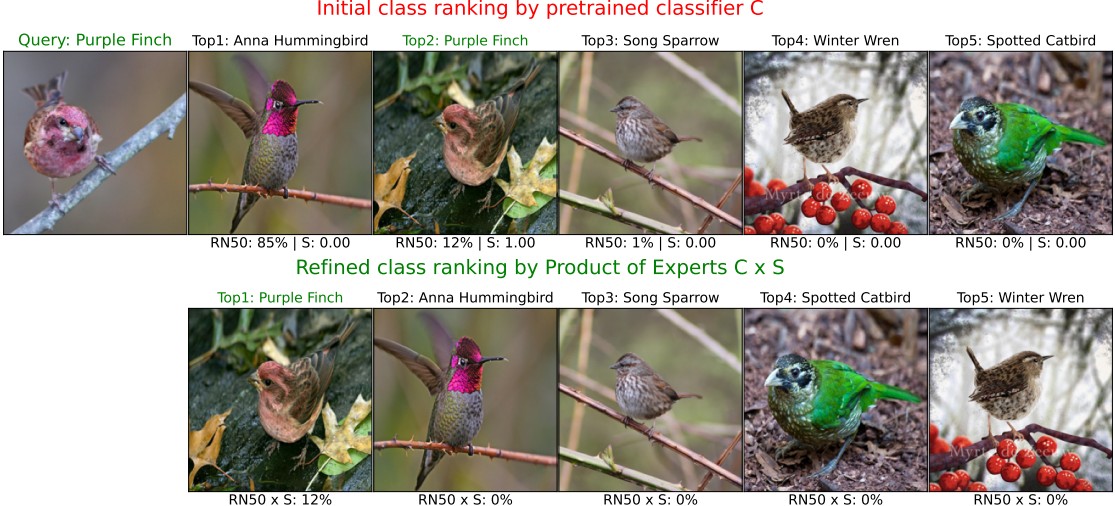

Figure 15: A pretrained model $\mathbf{C}$ makes predictions on the Query image (ground-truth label: Least Tern) and produces initial ranking (top row) but the top-1 predicted class (Forsters Tern) does not match the ground-truth label. Our classifier $\mathbf{C} \times \mathbf{S}$ compares the query image with the representative of each class (the first NN example in each class) and uses Product of Experts $\mathbf{C} \times \mathbf{S}$ to re-rank those classes. The refined class ranking is presented in the bottom row where the Least Tern class has been recognized as top-1.

Figure 16: A pretrained model $\mathbf{C}$ makes predictions on the Query image (ground-truth label: Purple Finch) and produces initial ranking (top row) but the top-1 predicted class (Anna Hummingbird) does not match the ground-truth label. The refined class ranking is presented in the bottom row where the Purple Finch class has been recognized as top-1.

Figure 17: A pretrained model **C** makes predictions on the Query image (ground-truth label: Lazuli Bunting) and produces initial ranking (top row) but the top-1 predicted class (Indigo Bunting) does not match the ground-truth label. The refined class ranking is presented in the bottom row where the Lazuli Bunting class has been recognized as top-1.

Figure 18: A pretrained model **C** makes predictions on the Query image (ground-truth label: Pine Warbler) and produces initial ranking (top row) but the top-1 predicted class (Yellow throated Vireo) does not match the ground-truth label. The refined class ranking is presented in the bottom row where the Pine Warbler class has been recognized as top-1.

**Initial class ranking by pretrained classifier C**

**Refined class ranking by Product of Experts C x S**

Figure 19: A pretrained model **C** makes predictions on the Query image (ground-truth label: Suzuki Aerio Sedan 2007) and produces initial ranking (top row) but the top-1 predicted class does not match the ground-truth label. Our classifier **C × S** compares the query image with the representative of each class (the first NN example in each class) to re-rank those classes based on confidence scores. The refined class ranking is presented in the bottom row where the Suzuki Aerio Sedan 2007 class has been recognized as top-1.

**Initial class ranking by pretrained classifier C**

**Refined class ranking by Product of Experts C x S**

Figure 20: A pretrained model **C** makes predictions on the Query image (ground-truth label: Nissan Leaf Hatchback 2012) and produces initial ranking (top row) but the top-1 predicted class does not match the ground-truth label. The refined class ranking is presented in the bottom row where the Nissan Leaf Hatchback 2012 class has been recognized as top-1.

**Initial class ranking by pretrained classifier C**

Figure 21: A pretrained model **C** makes predictions on the Query image (ground-truth label: Volvo XC90 SUV 2007) and produces initial ranking (top row) but the top-1 predicted class does not match the ground-truth label. The refined class ranking is presented in the bottom row where the Volvo XC90 SUV 2007 class has been recognized as top-1.

**Initial class ranking by pretrained classifier C**

Figure 22: A pretrained model **C** makes predictions on the Query image (ground-truth label: Ferrari FF Coupe 2012) and produces initial ranking (top row) but the top-1 predicted class does not match the ground-truth label. The refined class ranking is presented in the bottom row where the Ferrari FF Coupe 2012 class has been recognized as top-1.

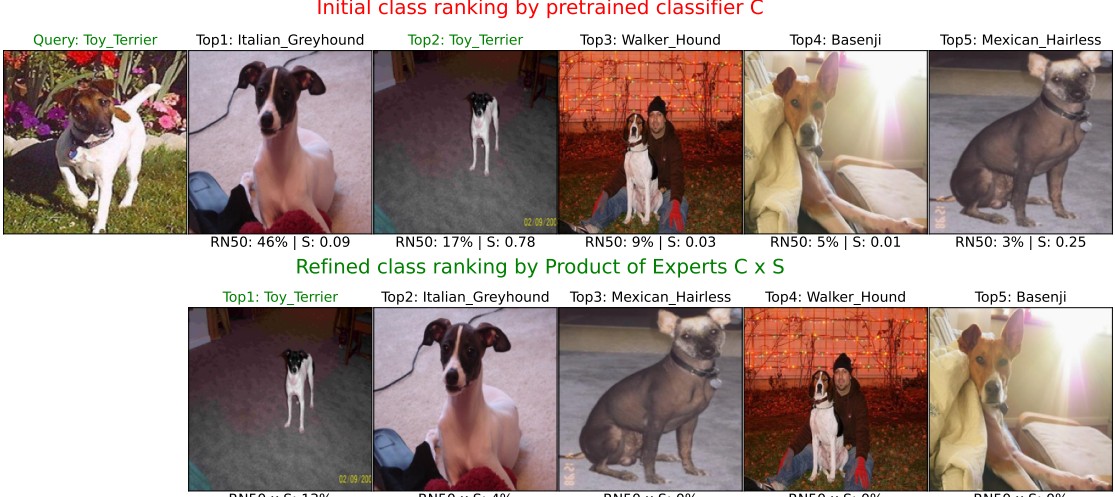

Initial class ranking by pretrained classifier C

Query: Border_Collie | Top1: Groenendael | Top2: Flat Coated_Retriever | Top3: Newfoundland | Top4: Border_Collie | Top5: Collie

RN50: 63% | S: 0.00 · RN50: 20% | S: 0.05 · RN50: 4% | S: 0.00 · RN50: 3% | S: 0.82 · RN50: 1% | S: 0.00

Refined class ranking by Product of Experts C x S

Top1: Border_Collie | Top2: Flat Coated_Retriever | Top3: Groenendael | Top4: Newfoundland | Top5: Collie

RN50 x S: 2% · RN50 x S: 1% · RN50 x S: 0% · RN50 x S: 0% · RN50 x S: 0%

Figure 23: A pretrained model **C** makes predictions on the Query image (ground-truth label: Border Collie) and produces initial ranking (top row) but the top-1 predicted class does not match the ground-truth label. Our classifier **C × S** compares the query image with the representative of each class (the first NN example in each class) to re-rank those classes based on confidence scores. The refined class ranking is presented in the bottom row where the Border Collie class has been recognized as top-1.

Initial class ranking by pretrained classifier C

Query: Toy_Terrier | Top1: Italian_Greyhound | Top2: Toy_Terrier | Top3: Walker_Hound | Top4: Basenji | Top5: Mexican_Hairless

RN50: 46% | S: 0.09 · RN50: 17% | S: 0.78 · RN50: 9% | S: 0.03 · RN50: 5% | S: 0.01 · RN50: 3% | S: 0.25

Refined class ranking by Product of Experts C x S

Top1: Toy_Terrier | Top2: Italian_Greyhound | Top3: Mexican_Hairless | Top4: Walker_Hound | Top5: Basenji

RN50 x S: 13% · RN50 x S: 4% · RN50 x S: 0% · RN50 x S: 0% · RN50 x S: 0%

Figure 24: A pretrained model **C** makes predictions on the Query image (ground-truth label: Toy Terrier) and produces initial ranking (top row) but the top-1 predicted class does not match the ground-truth label. The refined class ranking is presented in the bottom row where the Toy Terrier class has been recognized as top-1.

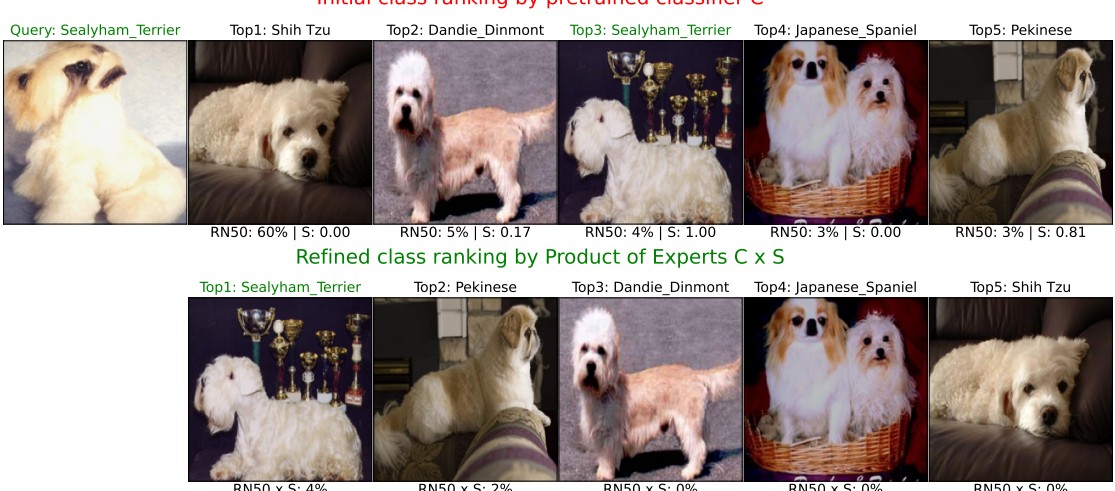

Figure 25: A pretrained model **C** makes predictions on the Query image (ground-truth label: `Otterhound`) and produces initial ranking (top row) but the top-1 predicted class does not match the ground-truth label. The refined class ranking is presented in the bottom row where the `Otterhound` class has been recognized as top-1.

Figure 26: A pretrained model **C** makes predictions on the Query image (ground-truth label: `Sealyham Terrier`) and produces initial ranking (top row) but the top-1 predicted class does not match the ground-truth label. The refined class ranking is presented in the bottom row where the `Sealyham Terrier` class has been recognized as top-1.

### I.2 Visualization of image comparator S's errors

The classification performance of **S** on **C**'s correct/incorrect predictions varies from 90% to 92% for both datasets as described in Sec. B.7. We are keen to investigate the cases where **S** does not perform accurately to better understand the limitations of them.

We find that if **S** **incorrectly** predicts that two images are from the same class, the image pairs often indeed appear to be very *similar*. Via this analysis, we are able to find multiple examples of two identical images which had two different labels (i.e., wrong annotations in Cars-196).

In contrast, when **S** **incorrectly** predicts that two images are *dissimilar*, the image pairs display notable differences (e.g., due to different lighting or angles). Qualitative results can be found in Fig. 27 for CUB-200 and Fig. 28 for Cars-196.

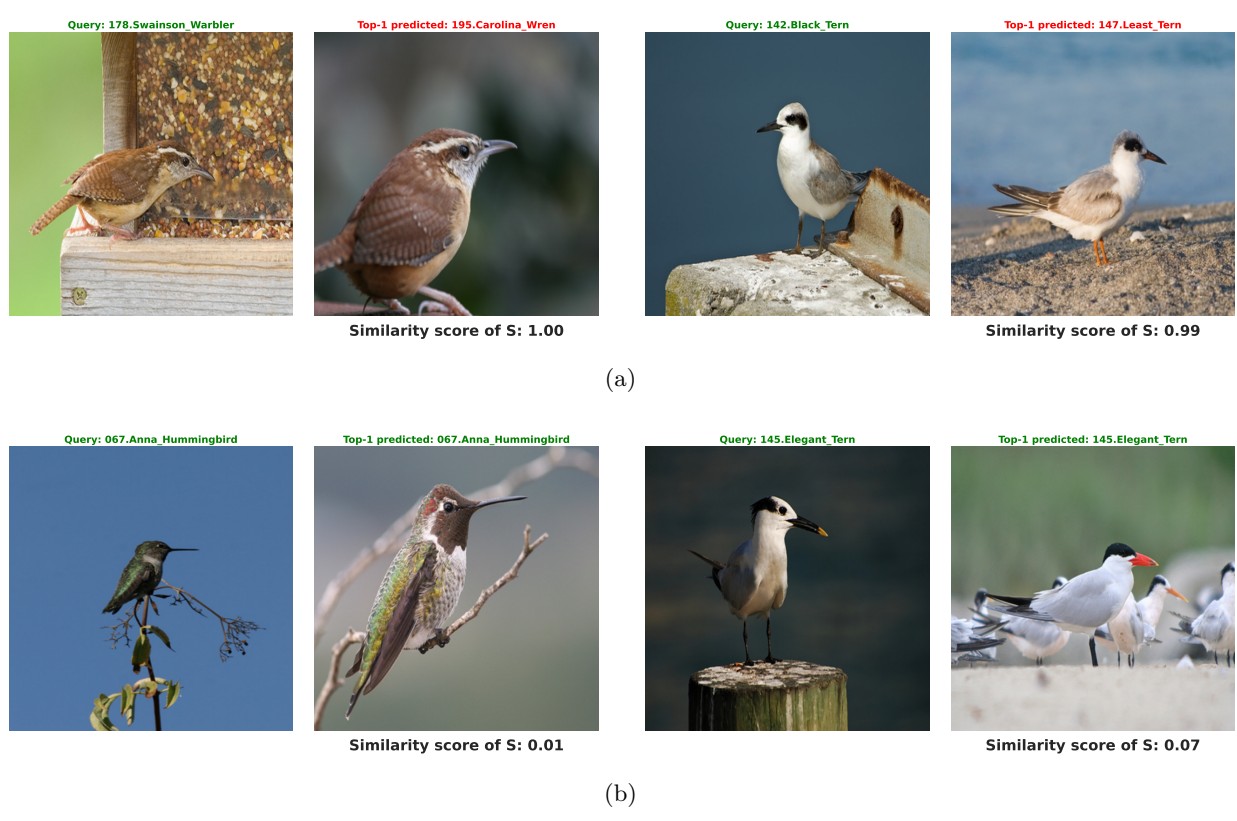

Figure 27: Failures of an image comparator **S** (94.44% test acc) on CUB-200. (a) **S** incorrectly accepted the **C**'s predictions for both images. For the left image, the two birds (Swainson Warbler vs. Carolina Wren) appear similar. For the right image, the birds (Black Tern vs. Least Tern) also appear visually similar. Both scenarios led **S** to accept with high confidence. (b) **S** incorrectly rejected the **C**'s predictions for both images. For the left image, the bird is a Anna Hummingbird. For the right image, the bird is a Elegant Tern. Both sets of birds show inter-class variations, leading **S** to mistakenly reject the predictions.

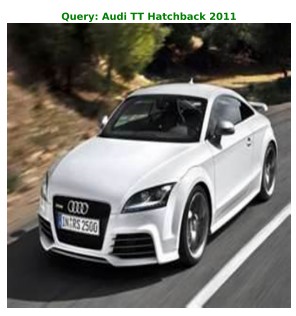
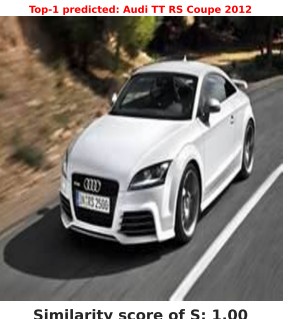
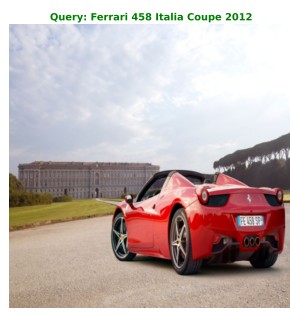
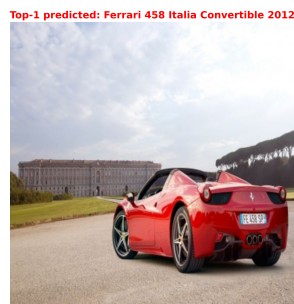

**Query: Audi TT Hatchback 2011** **Top-1 predicted: Audi TT RS Coupe 2012** **Query: Ferrari 458 Italia Coupe 2012** **Top-1 predicted: Ferrari 458 Italia Convertible 2012**

**Similarity score of S: 1.00** **Similarity score of S: 1.00**

(a)

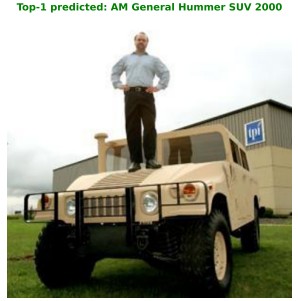
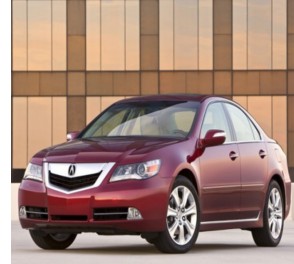
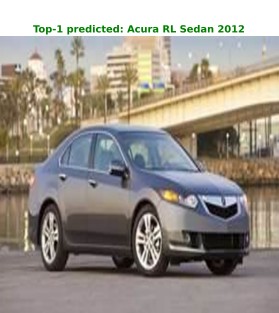

**Query: AM General Hummer SUV 2000** **Top-1 predicted: AM General Hummer SUV 2000** **Query: Acura RL Sedan 2012** **Top-1 predicted: Acura RL Sedan 2012**

**Similarity score of S: 0.00** **Similarity score of S: 0.47**

(b)

Figure 28: Failures of an image comparator **S** (95.02% test acc) on Cars-196. (a) **S** incorrectly accepted the **C**'s predictions for both images. We discovered that the cars are indeed identical. The image comparator's misclassifications are due to errors in Cars-196 dataset annotations. (b) **S** incorrectly rejected the **C**'s predictions for both images. We observed that, in most instances, the cars display noticeable differences, such as color, viewing angle, and lighting.

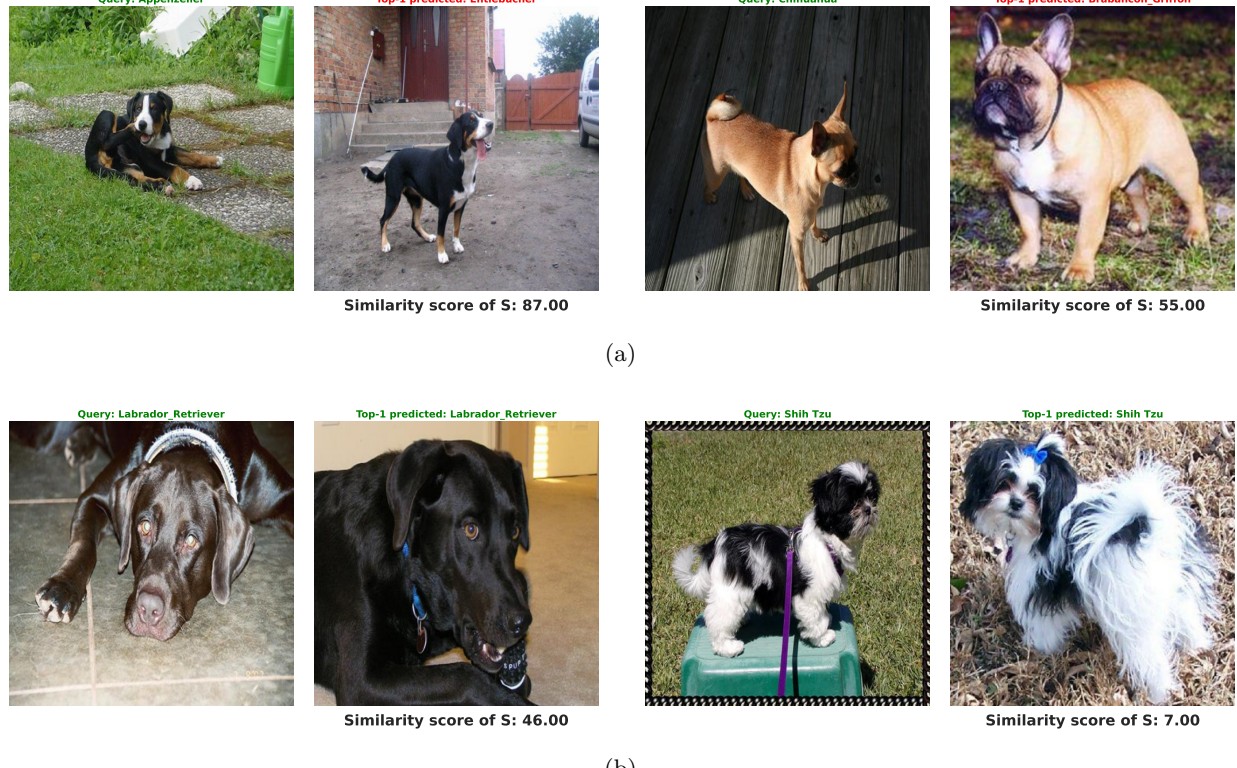

Figure 29: Failures of an image comparator **S** (92.05% test acc) on Dogs-120. (a) **S** incorrectly accepted the **C**'s predictions for both images. For the left image, the two dogs (Appenzeller vs. Entlebucher) appear similar. For the right image, the dogs (Chihuahua vs. Brabancon Griffon) also appear visually similar. Both scenarios led **S** to accept but only with low confidence. (b) **S** incorrectly rejected the **C**'s predictions for both images. For the left image, the dog is a Labrador Retriever. For the right image, the dog is a Shih Tzu. Both sets of dogs show variations in light and pose, leading **S** to mistakenly reject the predictions.

### I.3    Visualization of nearest neighbors $nn$

We show the nearest neighbors $nn$ utilized for forming training pairs in training **S** in Figs. 30 & 31 for CUB-200, Fig. 32 & 33 for Cars-196, and Fig. 34 & 35 for Dogs-120. From the top-1 predicted classes, we assemble $Q$ pairs. For the remaining classes beyond the top-1, we take the first nearest neighbor within each class and combine with the respective query sample. With $Q = 10$, a total of 19 pairs are returned by the sampling algorithm (Fig. 4).

When we prompt the pretrained model **C** to make predictions on its training data $x$, it is reasonable to expect the top-1 predicted class to match the ground-truth label. This expectation is grounded in the fact that model **C** typically exhibits nearly perfect accuracy (around 100%) on its training set.

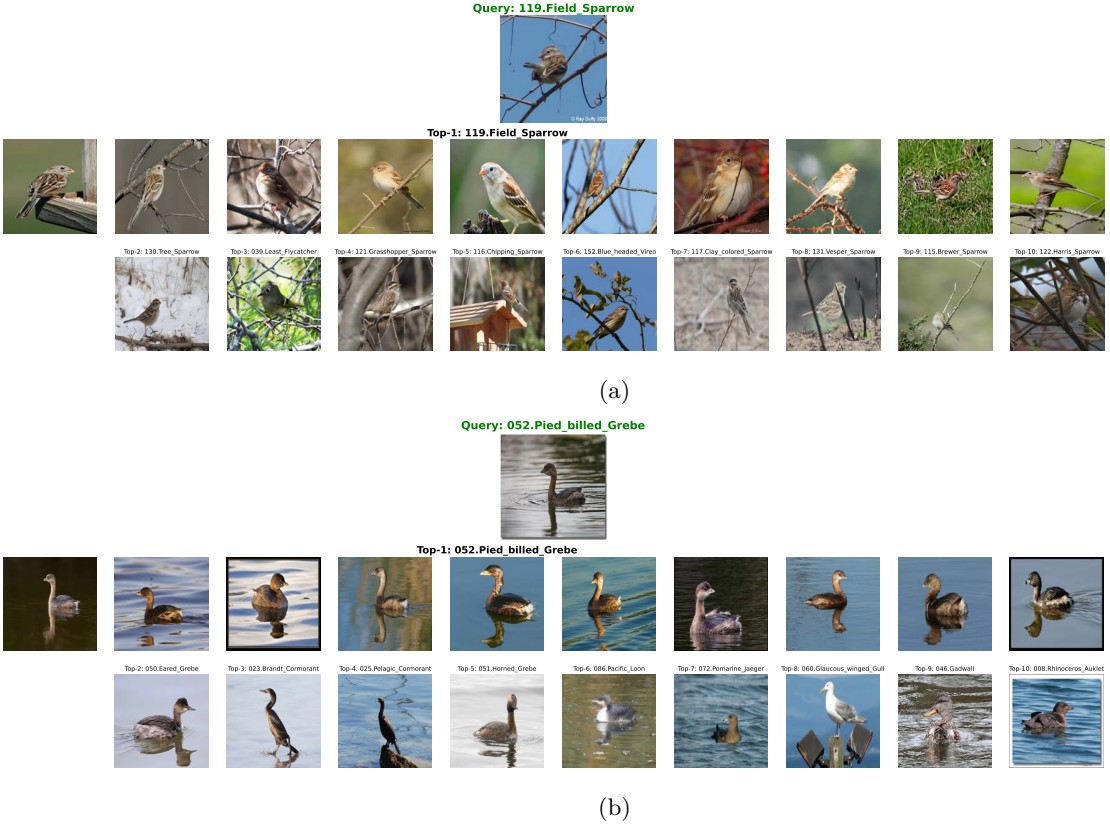

Figure 30: Training pairs for CUB-200 image comparator. For a given query, a pretrained model **C** predicts the label Field Sparrow (a) and Pied Billed Grebe (b), which align with the ground-truth labels. Subsequently, from each of these query images, we generate 10 positive pairs (using NNs of top-1 class in the middle row) and 9 negative pairs (using NNs of top-2 to 10 classes in the bottom row).

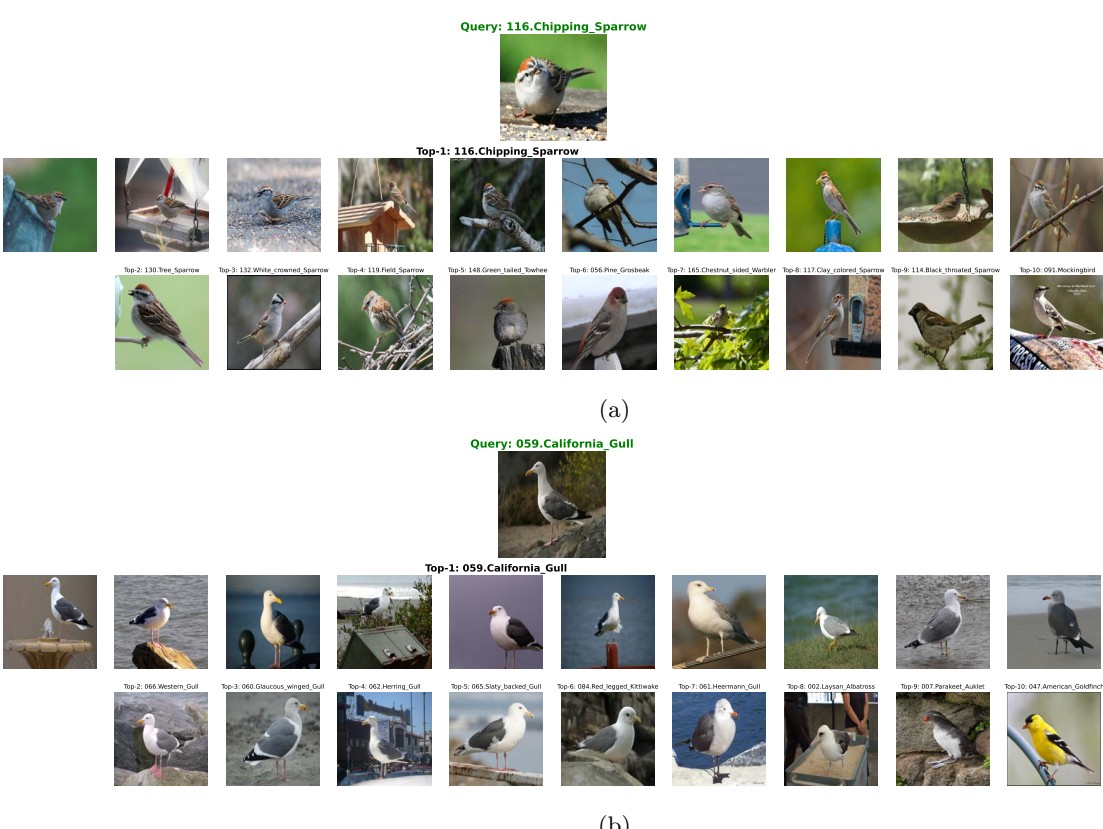

Figure 31: Training pairs for CUB-200 image comparator. For a given query, a pretrained model **C** predicts the label Chipping Sparrow (a) and California Gull (b), which align with the ground-truth labels. Subsequently, from each of these query images, we generate 10 positive pairs (using NNs of top-1 class in the middle row) and 9 negative pairs (using NNs of top-2 to 10 classes in the bottom row).

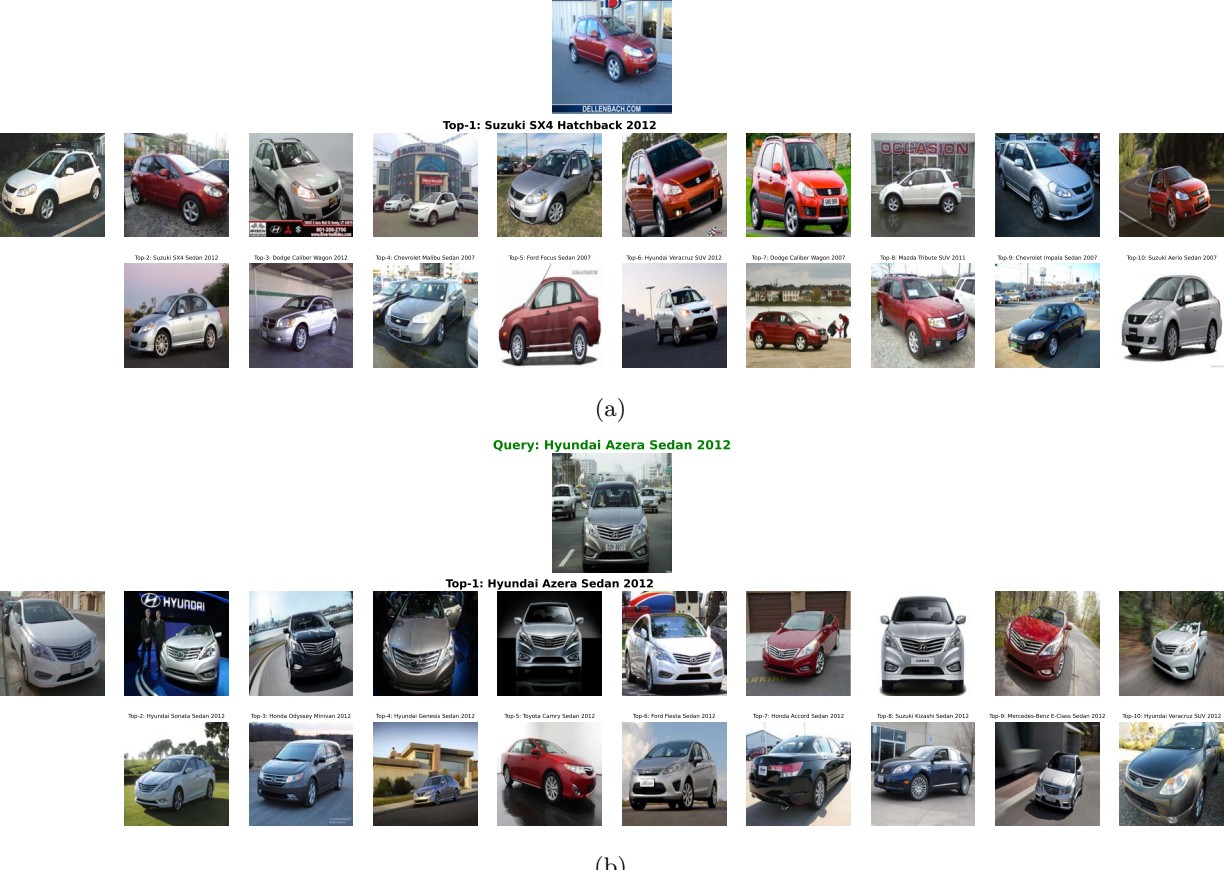

Figure 32: Training pairs for Cars-196 image comparator. For a given query, a pretrained model **C** predicts the label Suzuki SX4 Hatchback 2012 (a) and Hyundai Azera Sedan 2012 (b), which match the ground-truth labels. Subsequently, from each of these query images, we generate 10 positive pairs (using NNs of top-1 class in the middle row) and 9 negative pairs (using NNs of top-2 to 10 classes in the bottom row).

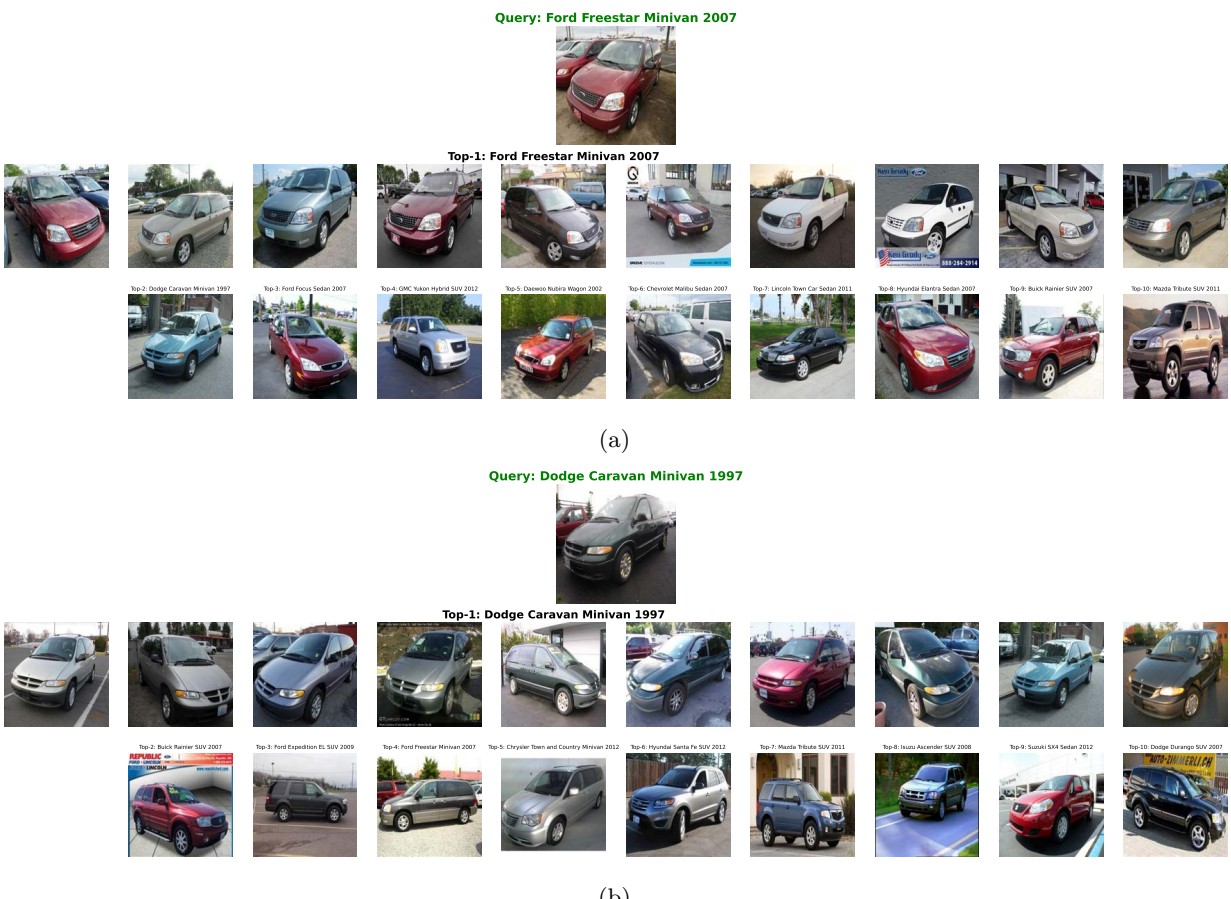

(a)

(b)

Figure 33: Training pairs for Cars-196 image comparator. For a given query, a pretrained model **C** predicts the label Ford Freestar Minivan 2007 (a) and Dodge Caravan Minivan 1997 (b), which match the ground-truth labels. Subsequently, from each of these query images, we generate 10 positive pairs (using NNs of top-1 class in the middle row) and 9 negative pairs (using NNs of top-2 to 10 classes in the bottom row).

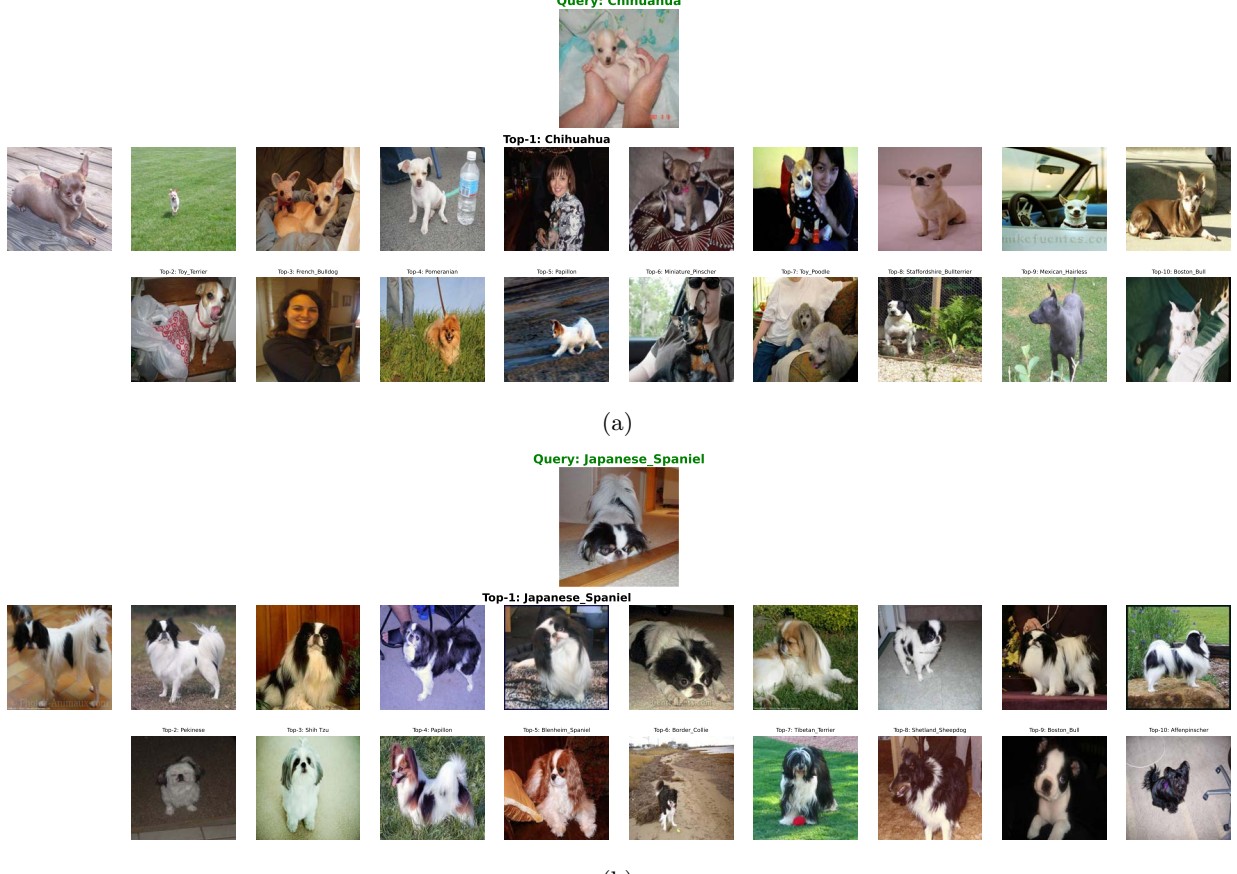

Figure 34: Training pairs for Dogs-120 image comparator. For a given query, a pretrained model **C** predicts the label Chihuahua (a) and Japanese Spaniel (b), which match the ground-truth labels. Subsequently, from each of these query images, we generate 10 positive pairs (using NNs of top-1 class in the middle row) and 9 negative pairs (using NNs of top-2 to 10 classes in the bottom row).

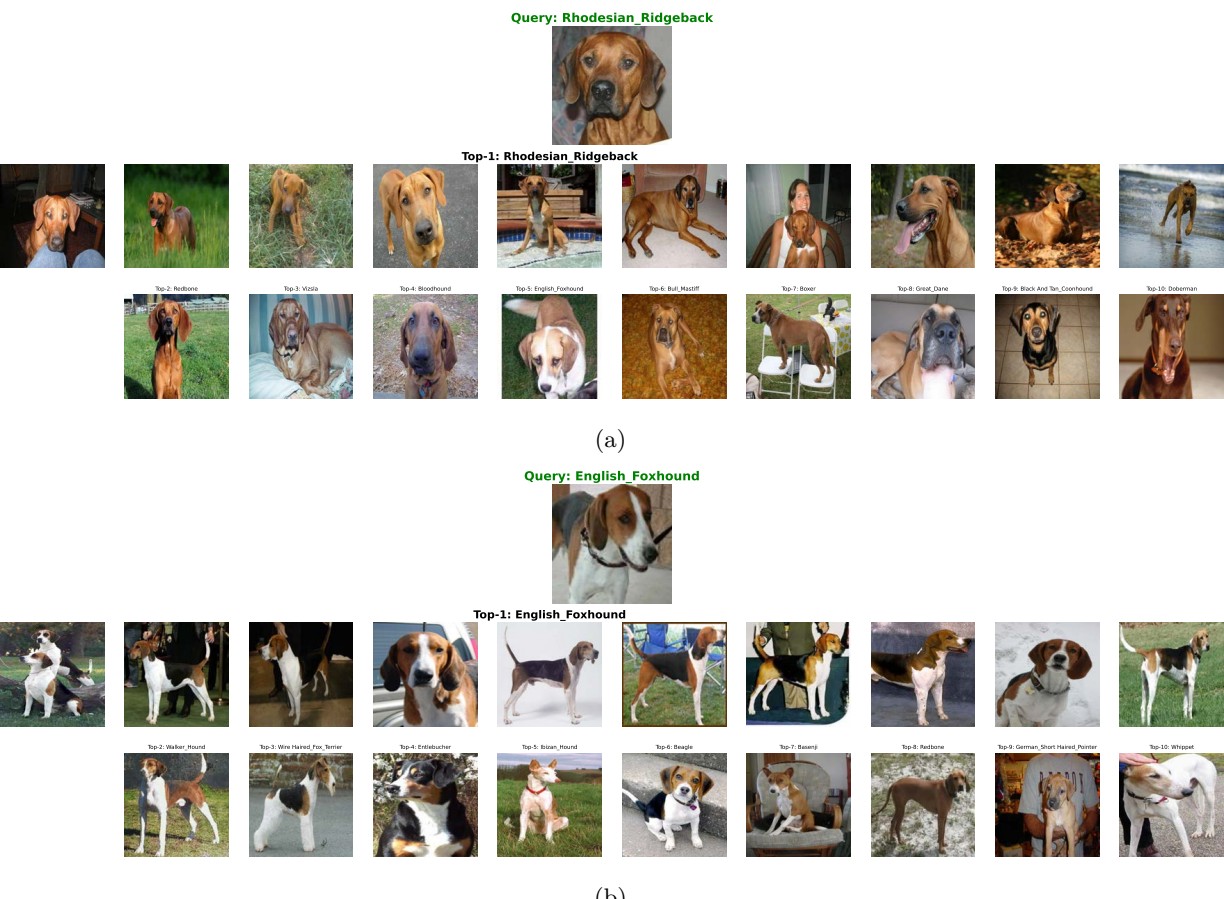

(a)

(b)

Figure 35: Training pairs for Dogs-120 image comparator. For a given query, a pretrained model **C** predicts the label Rhodesian Ridgeback (a) and English Foxhound (b), which match the ground-truth labels. Subsequently, from each of these query images, we generate 10 positive pairs (using NNs of top-1 class in the middle row) and 9 negative pairs (using NNs of top-2 to 10 classes in the bottom row).

