# OpenReview forum: "PCNN: Probable-Class Nearest-Neighbor Explanations Improve Fine-Grained Image Classification Accuracy for AIs and Humans"
_TMLR — Accepted by TMLR_

### Review · Reviewer_g7wm · 2024-04-27

**Summary Of Contributions:**

The paper proposes a novel method called "Probable-Class Nearest Neighbours (PCNN)" to improve the accuracy of fine-grained image classification models without retraining the base classifier.
PCNN creates an explanation set containing the nearest neighbour images from the top K predicted classes by the base classifier. These PCNN examples are used along with an image comparator model to re-rank the predictions of the base classifier.
The image comparator is a binary classifier trained to compare pairs of images and predict if they belong to the same class or not. It is trained on positive (same class) and "hard" negative (different but visually similar classes) image pairs sampled using the predictions of the base classifier.
The PCNN method improves classification accuracy across three fine-grained datasets (birds, cars, dogs) and multiple base classifier architectures like ResNets. It outperforms prior prototype-based explainable models.
A well-trained image comparator can be combined with arbitrary unseen base classifiers to boost their accuracy, without requiring retraining of the comparator.
In a human study, showing PCNN examples improved user accuracy by ~10% compared to just showing the top predicted class examples, which often misled users on visually similar categories.
The tradeoff is that the PCNN approach adds inference time overhead compared to just using the base classifier alone, due to the need to query the image comparator multiple times.

Summary of Contributions:

1. Proposed a novel explanation method called Probable-Class Nearest Neighbours (PCNN), which consists of K nearest images from the top-K predicted classes, to improve fine-grained image classification accuracy for both AI and humans.
2. Introduced a re-ranking algorithm that leverages an image comparator network (S) trained on PCNN to weigh and refine the predictions of a frozen, pretrained classifier (C), forming a Product of Experts (PoE) model.
3. Demonstrated consistent accuracy improvements of the PoE model over the original classifier C across three fine-grained image classification datasets (CUB-200, Cars-196, and Dogs-120) and multiple architectures (ResNet-18, ResNet-34, and ResNet-50).
4. Showed that a well-trained comparator S can be effectively combined with arbitrary, unseen classifiers C to boost their performance.
5. Outperformed state-of-the-art explainable and prototype-based classifiers on all three datasets.
6. Conducted a human study that showed that PCNN explanations improved users' decision-making accuracy compared to showing only top-1 class examples.

**Audience:**

Yes

**Claims And Evidence:**

Yes

**Requested Changes:**

Some high-level questions:

1. How does the proposed method perform on tasks other than fine-grained image classification, such as object detection or semantic segmentation (the paper is 47 pages long so hopefully I didn’t miss any results)?
2. Can the re-ranking algorithm be extended to handle multi-label classification problems?
3. How does the performance of the PoE model scale with the number of classes or the size of the dataset?
4. What is the impact of using different distance metrics or similarity measures for finding nearest neighbours on the performance of the PoE model?
5. Are there any theoretical insights or principles that can guide the design and training of the image comparator S?
6. In section 4.3 you say “First, we test using cosine similarity in the pretrained feature space of a CUB-200 classifier” – why do you call CUB-200 a classifier? CUB-200 is mentioned 80+ across the paper.

Potential Changes:

1. Explore techniques to reduce the computational overhead introduced by the re-ranking process, such as efficient nearest neighbour search or pruning strategies.
2. Extend the human study to other datasets and domains to further evaluate the interpretability and usefulness of PCNN explanations.
3. Provide a theoretical analysis or insights into the effectiveness of the proposed approach, potentially drawing connections to existing theories or principles in machine learning or cognitive science.


Lastly, it would be greatly appreciated if you have another go to fix typos and some unconventional syntax.
Also thanks for providing the source code.

**Strengths And Weaknesses:**

Strengths:

1. Novel and effective approach to leverage nearest neighbours for improving classification accuracy without retraining the original model.
2. Extensive experiments across multiple datasets and architectures, demonstrating the robustness and generalisability of the proposed method.
3. Thorough hyperparameter tuning and ablation studies to validate design choices.
4. Comprehensive comparisons with existing methods, showcasing the superior performance of the proposed approach.
5. Inclusion of a human study to evaluate the interpretability and usefulness of PCNN explanations for human decision-making.

It is worth highlighting that the authors have included several comprehensive appendixes showing some additional results and implementation details

-----------------------------------------------------------
As far as I can understand the nitty gritty of the method, the main weaknesses seem to be the following:

1. The re-ranking process introduces additional computational overhead, making the PoE model slower than the original classifier C.
2. The method relies on the availability of a well-trained comparator S, which may not be feasible or practical in certain scenarios.
3. The paper does not provide a theoretical analysis or justification for the effectiveness of the proposed approach.
4. The human study is limited to the CUB-200 dataset, and the generalizability of the findings to other domains is not explored.

---

> ### Author Response · Authors · 2024-05-18
> **Thank you and here is our rebuttal!**
>
> We sincerely thank you for your extremely thoughtful and constructive feedback!
>
> We are happy that your comments have helped us improved out paper's presentation substantially! We have not pushed a revision to OpenReview since we expect additional reviews and we wish to keep the same version for consistency among reviews.
>
> Please find our inline responses below.
>
> Updated: We incorporated our responses to your comments in the latest revision of our submission.

---

> ### Author Response · Authors · 2024-05-18
> **Regarding Question 1 & 2**
>
> ### **> (Question 1) How does the proposed method perform on tasks other than fine-grained image classification, such as object detection or semantic segmentation?**
>
> Thank you for a very interesting question! While our method is designed for image classification, PCNN's principles can be applied to object detection and segmentation.
>
> **Object detection.**  For example, in an input image, for each proposed bounding box `b` whose top-3 class labels are assigned similar probabilities, we can apply our method, i.e., re-rank the top-`K` predicted labels by comparing the embedding of the region of interest (RoI) `b` with that of the nearest training-set RoIs.
> However, the similarity function for retrieving the nearest RoI would need some research to answer some questions e.g. besides the content inside the RoI, should we consider its location in the image, dimensions of the RoI, and the context outside the RoI?
> For **segmentation**, the shape of the predicted segmentation may need also to be considered.
>
> We will discuss this idea for future work in our paper!
> ***
>
> ### **> (Question 2) Can the re-ranking algorithm be extended to handle multi-label classification problems?**
>
> ***Yes, it can!*** First, let's revisit the definition of multi-label classification for clarity.
>
> Multi-label classification is a type of machine learning problem where each instance (or example) can
> be assigned multiple labels from a set of possible labels.
> This differs from the traditional single-label classification tasks, where each instance is
> associated with only one label.
>
> Let's take an example from C-Tran paper [a].
> In their [Figure](https://drive.google.com/file/d/1ltlCz-pIVm1e7s8133XMeCGWfkgP-aRn/view?usp=sharing) [a], the based classifier are predicting the presence of a `person` and an `umbrella` in the input image.
> The threshold to determine presence is set to 0.5 (specified in `Sec. 4.1` of C-Tran paper [a]).
>
> In multi-label classification, the ground-truth y is a binary vectors indicating the presence of C classes `{y1, y2, ..., yc}, yi ∈ {0, 1}`.
>
> Then, for each label, the two classes to be re-ranked here are the presence (1) and absence (0) of the object (e.g. `person`).
> Extending our method to multi-label classification, we'd need to:
> 1. Construct two training-set sub-classes -- presence and absence -- for each class (e.g. person vs. no person).
> 2. Apply our re-ranking to these two sub-classes (i.e. `K = 2` in this case).
>
> Because our re-ranking steps add extra run-time to the classification pipeline, we'd perform the above two steps for predicted labels whose confidence scores are in the **unsure** range (i.e. around the cut-off 0.5 threshold, e.g., 0.35 - 0.65), which is a hyperparameter that needs to be tuned.
> Based on the single-label fine-grained classification results in the paper, we believe that because our method leverages training-set examples and a comparator `S`, our PoE re-ranking can also help improve the accuracy on top of a multi-label classifier `C` when `C` is unsure.
>
> ################## References ##################
>
> - [a] [General Multi-label Image Classification with Transformers](https://openaccess.thecvf.com/content/CVPR2021/papers/Lanchantin_General_Multi-Label_Image_Classification_With_Transformers_CVPR_2021_paper.pdf), CVPR 2021

---

> ### Author Response · Authors · 2024-05-18
> **Regarding Question 3**
>
> ### **> (Question 3) How does the performance of the PoE model scale with the number of classes or the size of the dataset?**
>
> We assume that, by "performance", the reviewer means **test-set accuracy** of the PoE model.
>
> Our intuition is that, given a challenging fine-grained classification task and a classifier C that is _suboptimal_, our PoE method improves classification further by reviewing the training-set images and using an external image comparator S. Our [Fig. 6](https://drive.google.com/file/d/1IU-_sjf_uUgOcWHIOt-aBuBaVAyMnqT1/view?usp=sharing) shows a trend that the less accurate the classifier C, the larger improvement PoE will bring.
>
> We agree that varying the number of classes or the number of images can change the accuracy of classifier C or comparator S. In turn, these components will affect the accuracy of the PoE CxS model.
>
> **(a) Changing the number of classes**
>
> First, in the context of fine-grained classification, if we increase the number of classes substantially (e.g. from 200 to 10,000) assuming a class-balance dataset, we would expect the classifier C to be more discriminative and since it would be trained on a better dataset (more labels and more discriminative supervision signals).
> In that case, C would benefit less from our method (i.e. PoE and C may perform similarly).
>
> In contrast, if C is trained and tested on a 200-class dataset, but S is trained on a 10K-class training set, S would benefit from this larger dataset and therefore should improve the PoE accuracy further.
>
> If C and S are already trained and frozen, just reducing the number of classes at the evaluation time should make the task easier and therefore, the benefit from PoE would be less.
>
> **(b) Increasing the size of the dataset**
>
> There are many ways to interpret this scenario but the main impact this has to PoE accuracy boils down to whether this dataset-size upscaling would improve the accuracy of C or S.
> That is, if a larger dataset improves C accuracy substantially on the same task, then, there is less benefit of applying PoE.
> In contrast, for a fixed, frozen C, if upscaling the dataset leads to a better trained S, the accuracy of PoE should increase.
>
> In sum, based on your questions, we believe [Fig. 6](https://drive.google.com/file/d/1IU-_sjf_uUgOcWHIOt-aBuBaVAyMnqT1/view?usp=sharing) in the paper and our explanation here have addressed your scaling question. Yet, if you have any specific experiments in mind, we are happy to run and report in the paper!

---

> ### Author Response · Authors · 2024-05-18
> **Regarding Question 4**
>
> ### **> (Question 4) What is the impact of using different distance metrics or similarity measures for finding nearest neighbours (NNs) on the performance of the PoE model?**
>
> Thank you for the insightful question! We want to break down the answer into two parts:
>
> **Part 4a**. When the NNs are already found using `L2` distance via the `IndexFlatL2` function of `faiss` library [1],
> we tested the impact of using different functions to provide a similarity score (between the input image and each NN) in the re-ranking algorithm.
>
>
> | NN-th |  |        | Top-1 Acc on CUB-200      |||
> |-------|----------|-----------------|--------|------|------------------|
> |       | (a) C × S | (b) Our trained S** | cosine | EMD  | 4-layer MLP only |
> | 1st   | 88.59    | 87.72           | 60.20  | 54.83 | 83.76            |
> | 2nd   | 88.06    | 87.34           | 58.84  | 57.05 | 84.33            |
> | 3rd   | 88.21    | 87.43           | 57.47  | 57.14 | 83.93            |
>
> ** S = shared feature extractor + 4-layer MLP for similarity function
>
> See `Tab. 7` in our paper (also shown above) shows that the similarity scores returned by our trained S result in substantially better  C -> S accuracy (i.e., using only S scores to re-rank C's predicted label) than other similarity scores (cosine, EMD, and a 4-layer MLP).
>
> Using the CxS scores (here, C is a ResNet-50 on CUB-200) further improve the classification accuracy (a vs. b) compared to using S alone (b).
>
> ---
> **Part 4b.** As the reviewer requested, we fix the re-ranking method to PoE (C × S) and test using different distance metrics to find the nearest neighbors.
>
> Here we assess compare the PoE accuracy when NNs are retrieved using `cosine` similarity vs. DreamSim [2] vs `L2`.
> `cosine` similarity measure the image-level similarity between two images while `DreamSim` [2] is a human-aligned similarity measure.
>
> As shown in Table below, we find that using these measures have a small to no impact on the PoE model performance.
>
>
> | Distance function                              | CUB-200 (%) | Dogs-120 (%) |
> |------------------------------------------------|----------------------|-----------------------|
> | Random                                         | 87.95%               | 85.89%                |
> | L2                                             | 88.59%               | 86.31%                |
> | Cosine                                         | 88.33%               | 86.25%                |
> | DreamSim[2] | 88.38%               | 86.56%                |
>
> **_The top-1 accuracy of PoE on CUB-200 and Dogs-120 over different choices of distance functions_**
>
> We suppose that because the number of samples per class is fairly small (`30` images per class on average) when doing retrieval with different distance metrics,
> the nearest neighbors are **not very different**.
>
> Then, we move from CUB-200 to Dogs-120 that has `3x` more samples per class (`100` images per class on average).
> Interestingly, different distance metrics also have a marginal impact on the PoE model performance.
>
> We also test using the **Random nearest neighbors** (i.e. randomly picking one sample from each of the probable classes for re-ranking). Yet, it consistently drops the PoE model performance from `88.59` to `87.95` on CUB-200 and from `86.31` to `85.89` on Dogs-120, respectively.
>
> Overall, **we found that changing the distance metrics for finding nearest neighbors has a small to no impact on the performance of the PoE model**.
> In addition, the Random baseline confirms the robust of our re-ranking method based on class-wise NNs (i.e. taking 1 NN per class).
>
> ################## References ##################
>
> - [1] [Billion-scale similarity search with GPUs](https://arxiv.org/pdf/1702.08734), IEEE Transactions on Big Data 2019
> - [2] [DreamSim: Learning New Dimensions of Human Visual Similarity using Synthetic Data](https://arxiv.org/pdf/2306.09344), NeurIPS 2023

---

> ### Author Response · Authors · 2024-05-18
> **Regarding Question 5 & 6**
>
> ### **> (Question 5) Are there any theoretical insights or principles that can guide the design and training of the image comparator S?**
>
> Our work is inspired by many principles in computer vision and machine learning. Below we list out the insights we leveraged for designing the architecture and training of S.
>
> 1. **The task of the image comparator** S is inspired by the distinction task [1] where we give human users two images
> and ask them to judge if the two images are of the same class or not, which is a simpler task than identifying one label among `N` labels.
> 2. **To compare the two images**, we are inspired by [Siamese neural networks](https://drive.google.com/file/d/1xXblL8rHRqz1FWr6ulDbo0gKgKoP7yBG/view?usp=sharing) [2] where the feature extractor
> is shared between two branches and the similarity function is learnable. In the experiment in `Tab. 7` in our submission, we find that the proposed architecture (shared feature extractor + 4-layer MLP for similarity function) works the best for PoE model.
>
>
> | NN-th | Top-1 Acc | Top-1 Acc       | Top-1 Acc       | Top-1 Acc     | Top-1 Acc        |
> |-------|----------|-----------------|--------|------|------------------|
> |       | RN50 × S | Our trained S** | cosine | EMD  | 4-layer MLP only |
> | 1st   | 88.59    | 87.72           | 60.20  | 54.83 | 83.76            |
> | 2nd   | 88.06    | 87.34           | 58.84  | 57.05 | 84.33            |
> | 3rd   | 88.21    | 87.43           | 57.47  | 57.14 | 83.93            |
>
> ** S = shared feature extractor + 4-layer MLP for similarity function
>
> 3. **In training of S**, we are inspired by a large body of work in contrastive learning [3]. Consistent with the literature, we find that easy negatives (i.e. constructing a negative pair by taking two images from two random classes **is not optimal**) as shown in  Appendix `Sec. B.3` in the submission. Instead, we propose a novel hard-negative sampling based on a pretrained classifier, here, pairing up samples from a probable-class that is not the groundtruth class with one from the groundtruth class.
> The change from random negative-sampling to this proposed hard-negative sampling improves the PoE accuracy `86.55%` → `88.59%` on CUB-200 (see Appendix `Sec. B.3`).
>
> ---
> ### **> (Question 6) In section 4.3 you say “First, we test using cosine similarity in the pretrained feature space of a CUB-200 classifier” – why do you call CUB-200 a classifier? CUB-200 is mentioned 80+ across the paper.**
>
> Thank you for pointing this out! We felt the need to write "CUB-200 classifier" since there are three different datasets in the paper (CUB-200, Dogs-120, and Cars-196) and the readers may not know which classifier we are referring to.
>
> We will revise the writing to define this convention upfront i.e. `a classifier trained on CUB-200 (hereafter, CUB-200 classifier)` to avoid any misunderstanding.
>
> ################## References ##################
>
> - [1] [HIVE: Evaluating the Human Interpretability of Visual Explanations](https://arxiv.org/pdf/2112.03184), ECCV 2022
> - [2] [Siamese Neural Networks for One-shot Image Recognition](https://www.cs.cmu.edu/~rsalakhu/papers/oneshot1.pdf), ICML 2015
> - [3] [A Simple Framework for Contrastive Learning of Visual Representations](https://arxiv.org/pdf/2002.05709), ICML 2020

---

> ### Author Response · Authors · 2024-05-18
> **Regarding Potential Change 1**
>
> ### **> (Potential Change 1) Explore techniques to reduce the computational overhead introduced by the re-ranking process, such as efficient nearest neighbour search or pruning strategies.**
>
>
> Thanks for bringing up this point, and it really encourages us to optimize the runtime of our method!
>
> We agree that the re-ranking process can be computationally expensive, especially when the dataset is large.
> The computational overhead comes from (a) finding the nearest neighbors or (b) querying the image comparator S for re-ranking weights as shown in `Table. 11` in PCNN submission.
>
> (a) **nearest neighbor retrieval** can be already sped up by leveraging the speedup options of as listed [here](https://github.com/facebookresearch/faiss/wiki/How-to-make-Faiss-run-faster).
> For example, as we are currently using CPU for faiss indexing (`L137` in the submitted [code](https://anonymous.4open.science/r/nearest-neighbor-XAI-FF2E/cub-200/cub_extract_feature_for_reranking.py)),
> using GPU by `GpuIndexFlatL2` from faiss can provide significant speedups, especially for larger datasets.
>
> (b) We can reduce the number of queries to the image comparator S by ignoring less probable labels.
> Currently, we are always examining the top-`K` (with `K=10`) most probable classes.
> Yet, there always exists classes assigned a `< 1%` probability by ResNet50 (`Fig. 5` in PCNN submission) and are not in the top-1 after re-ranking.
> Reducing the number of `K` can save a lot of computation at a minimal cost of accuracy.
>
> To verify this, we run an experiment for CUB-200 where we instead of re-ranking the whole **top-10**, we only re-rank the classes that have a probability `>= 1%` assigned by the base classifier C.
> We found that:
> - The CxS model accuracy on CUB-200 drops very marginally by `only 0.08%` (from `88.59%` → `88.51%`).
> - However, the number of queries to the image comparator S was reduced by approx. `4x` (from `10` to just about `2.5` queries/image).
> This leads to a `2.5x` speedup in the overall runtime of the CxS system (from `64.55` seconds to `28.95` seconds per 1000 images), as shown in the following Table.
>
> | Model                        |   Time (s)   | Top-1 Acc (%) |
> |:----------------------------:|:------------:|:-------------:|
> | RN50 xS (before)                      | 64.55 ± 0.35 |     88.59     |
> | RN50 xS (after) | 28.95 ± 0.11 |     88.51     |
>
> **_The run-time of CxS on 1,000 queries on one Nvidia V100 GPU._**

---

> ### Author Response · Authors · 2024-05-18
> **Regarding Potential Change 2**
>
> ### **> (Potential Change 2) Extend the human study to other datasets and domains to further evaluate the interpretability and usefulness of PCNN explanations.**
>
> We agree!
> Per your request, we **repeat** the human study that compares top-1 nearest neighbors and PCNN on `Stanford Dogs-120` dataset.
>
> - Samples: Similar to CUB-200 study detailed in `Sec. H` in PCNN submission, we select 300 correctly classified and 300
> misclassified query samples determined by the RN50 × S classifier, amounting to a total of 600 images for the study.
>
> - Participants: We recruit `30` participants for the study, with `16` participants for the **top-1** and `14` participants for **PCNN** experiments.
> The participants are encouraged to perform responsibly, and we consider only the data from those who complete all the 30 test trials.
>
> - Findings: We report the results from the human study in this [Figure](https://drive.google.com/file/d/1Fh3GrO6blL3muijc2-hGTnp-NwCUoYAU/view?usp=sharing) and below Table:
>
> | Explanation | AI Correctness | mean (%) | std (%) | Numb. of Samples |
> |------------|----------------|----------|---------|------------------|
> | Top-1      | AI is Wrong    | 34.86    | 24.46   | 225              |
> | Top-1      | AI is Correct  | 89.07    | 9.10    | 255              |
> | Overall    | ---            | 63.66    | 27.05   | 480              |
> | PCNN       | AI is Wrong    | 52.74    | 18.63   | 192              |
> | PCNN       | AI is Correct  | 82.55    | 9.80    | 228              |
> | Overall    | ---            | **68.92**    | **14.85**   | 420              |
> **_Human accuracy on Stanford Dogs-120_**
>
>
> Finding: **Presenting humans with PCNN explanations improves their mean accuracy** (over presenting top-1 nearest images)  +`5.26` on Stanford Dogs, from `63.66%` to `68.92%`.
>
> When the AI is correct, participants achieve a lower mean accuracy of `82.55%` (± 9.80) with PCNN explanations, compared to `89.07%` (± 9.10) with top-1 nearest neighbors.
> However, when the AI is incorrect, participants with PCNN explanations achieve a significantly higher mean accuracy of `52.74%` (± 18.63) vs. `34.86%` (± 24.46) accuracy of those with top-1 nearest neighbors.
> These findings suggest that PCNN explanations provide more informative cues for humans to recognize the correctness of AI predictions, particularly in cases where the AI is incorrect.
> The results from the Stanford Dogs-120 dataset corroborate the findings from the CUB-200 study, further demonstrating the usefulness of PCNN explanations across different domains.
>
> The reviewer can try the human study on Stanford Dogs-120 via this [link](https://huggingface.co/spaces/xairesearch2023-advnet/HumanStudy-Dogs).

---

> ### Author Response · Authors · 2024-05-18
> **Regarding Potential Change 3**
>
> > (Potential Change 3) Provide a theoretical analysis or insights into the effectiveness of the proposed approach, potentially drawing connections to existing theories or principles in machine learning or cognitive science.
>
> Please see our response for <[Question 5](https://openreview.net/forum?id=OcFjqiJ98b&noteId=S8do3VQoSU)>.

---

> ### Author Response · Authors · 2024-05-18
> **Regarding Weakness 2**
>
> ### **> (Weakness 2) The method relies on the availability of a well-trained comparator S, which may not be feasible or practical in certain scenarios.**
>
> Our method has been proven to work very well on small-size, fine-grained datasets (e.g. CUB-200 or Dogs-120) where the comparator S can be trained successfully.
>
> For large-scale, general domains, one can leverage existing state-of-the-art image similarity metrics like DreamSim [1] or DINO [2] that were trained on large-scale and contrastive data.
> In the `Related Work` section of DreamSim paper [1], you can find several other alternative metrics for general domains (e.g. ImageNet).
> Adapting these metrics in our PoE model is straightforward as what we did with `cosine` or `Earth Mover's Distance` [3] in `Tab. 7` in the paper.
>
> ################## References ##################
>
> - [1] [DreamSim: Learning New Dimensions of Human Visual Similarity using Synthetic Data](https://arxiv.org/pdf/2306.09344), NeurIPS 2023
> - [2] [Emerging Properties in Self-Supervised Vision Transformers](https://arxiv.org/abs/2104.14294), CVPR 2021
> - [3] [Visual correspondence-based explanations improve AI robustness and human-AI team accuracy](https://arxiv.org/abs/2208.00780), NeurIPS 2022

---

> ### Author Response · Authors · 2024-05-18
> **Regarding Weakness 3 & 4**
>
> For Weakness 3, please see our response for <[Question 5](https://openreview.net/forum?id=OcFjqiJ98b&noteId=S8do3VQoSU)>.
>
> For Weakness 4, please see our response for <[Potential Change 2](https://openreview.net/forum?id=OcFjqiJ98b&noteId=lDRCZ95WQy)>.

---

> ### Comment · Reviewer_g7wm · 2024-07-05
> **Thank you**
>
> thank you for the very detailed rebuttal. I am happy with the responses as well as the modifications made to the paper.

---

> ### Author Response · Authors · 2024-07-08
>
> Thank you!
>
> We appreciate your suggestions and will make sure to include the discussion here into the final version of our paper.

---

### Review · Reviewer_bLsM · 2024-06-10

**Summary Of Contributions:**

The authors consider using nearest neighbor explanations (NNEs) of an image
classifier's top-K classes for improving (1) human decision making accuracy,
and (2) the model's accuracy. The second point is studied using a pipeline in
which the NNEs are used for reweighting/reranking the model's predictions
themselves, denoted PCNN.  Reranking is implemented by multiplying together the
probability that the input lies in a class y with the probability that the 1-NN
example belongs to the same class.  This is predicted by a CrossViT-based
``comparator'' fed with the input and the 1-NN.  Point (1) is motivated by the
fact that showing users only the 1-NN explanation is potentially limiting, and
in fact can hinder human performance in user studies.  Point (2) stems from a
desire to apply a similar logic to machine decisions.  The results highlight
how PCNN improves machine performance over various competitors on three
fine-grained image classification data and human performance on CUB-200.

**Audience:**

Yes

**Broader Impact Concerns:**

Broader impact should briefly discuss the potential impact of manipulating (in the case of PCNN, lowering) user confidence in machine predictions in, especially in **time critical** high-stakes scenarios.

**Claims And Evidence:**

No

**Requested Changes:**

- The authors are upfront about the fact that PCNN requires training data at
  test time (like all kNN-based predictors), but do not list this as an actual
  limitation in Section 5, while I think it is.

  There they mention run-time of PCNN is longer than other competitors, but the
  main issue is PCNN requires the training data to be available in entirety --
  or, at least, the experiments do not study the impact of reducing the sice of
  the training data on inference-time performance.  This is a clear downside
  compared to, say, ProtoPNets, which memorize relevant (part) prototypes
  instead.

  So there exists a clear trade-off between space and time requirements and
  prediction improvements (which is substantial but not huge to begin with,
  usually in the order of 2-3% top-1 accuracy over the runner-ups, at least
  according to Tables 3-5).

- I don't think the results of the user study are reported appropriately in the
  introduction.

  Bottom line: showing more NNs to users makes them more skeptical, meaning
  they end up underestimating machine perforance.  This should be clearly
  stated in the introduction, at the bare minimum.  Instead, the authors focus
  on the benefits of PCNN only, and write: "A 60-user study finds that,
  compared to showing top-1 class examples, PCNN improves user performance on
  the distinction task by almost +10 points (pp) (54.55% vs. 64.58%) on CUB-200
  (Sec. 4.6)."  I don't think this is entirely fair and the text should be
  amended.  This should also be listed in the Limitations section.

- The construction of the training set for S assumes C is already reasonably
  high-quality: is this always a reasonable assumption?  Please add this to the
  Limitations section too.

- An analysis of errors introduced by the reranking step would have been useful.

- p 4: "We empirically test K = {1, 3, 5, 10} and find K = 10 to be optimal."
  The fact that the optimal value is at the very end of the spectrum begs the
  question whether increasing K could improve performance further.  Did the
  authors evaluate what happens for larger values of K?  Clearly, increasing K
  would not be ideal for human decision making, but it should not consistute
  for PCNNs proper.

- p 2 onwards: The authors say their model is a Product of Experts (PoE), based
  on the definition of Hinton, 1999.  Reading through this reference, however,
  I get the impression that in PoEs the various distributions are conditionally
  independent given the input (for instance, in p 4 of Hinton '99, they state
  "the hidden states of different experts are conditionally independent given
  the data").  The same (conditional) independence assumption seems to be
  instrumental in more recent research on PoEs, see:

    Gordon et al., "Identifiability of Product of Experts Models." 2024.

  To me, conditional independence seems necessary to reinterpret the product of
  distributions as a factorization of a more complex joint distribution, which
  lies at the heart of PoEs.

  However, independence does not appear to hold for the CSP model.  I would
  appreciate if the authors could clarify this point, and -- if independence is
  indeed a prerequisite of PoEs -- changed their wording accordingly.

I am willing to change "Claims and Evidence" to "Yes" if the authors expressly discuss these limitations in the relevant section.

**Strengths And Weaknesses:**

Strengths

**Presentation**: Well written and structured, easy to follow.  All figures
are readable.  I did not find any major linguistic issues.

**Novelty**:  PCNN is, to the best of my knowledge, novel.  It shares many
similarities to existing kNN and prototype-based approaches, but it is overall
original.

**Significance**:  PCNN appears to attain better classification performance
than existing data sets on a challenging data set (CUB-200).  The idea of

**Quality**:  The proposed technique (PCNN) is rather simple but also
technically sound.  The experimental setup of the machine experiments is rather
exhaustive and indicates PCNN can lead to better machine performance.  The user
study seems to be setup appropriately (but it should probably be double checked
for confounding by an expert, e.g., a behavioral psychologist).

**Related Work**: Coverage of related works is good and these are portrayed
fairly.


Weaknesses

**Presentation**:  The message comprises two disconnected pieces: 1) showing
explanations for the top-K classes aids human decision making, 2) reranking
predictions using NNs helps machine performance.  This is not a big deal, but a
more streamlined message would have helped.  Another issue is the method name
is not used consistently throughout the text.  For instance, it is sometimes
called C \times S, sometimes PoE; sometimes PCNN is an architecture, sometimes
a new type of explanation; this is a bit confusing.  I would prefer if the
authors used PCNN everywhere, for simplicity.

**Quality**:  I don't think the paper is entirely open about all its
limitations, see below.




**TL;DR**: okay paper that builds on a nice idea, but that also requires more
  emphasis on limitations.

On the plus side, I think the paper is well written and worth accepting.
  I like the main intuition for the user study -- namely that providing
  (almost) counterfactuals to users can lead to more careful decision
  making -- although I don't think this is entirely in line with the
  experimental results.  The proposed AI pipeline is sensible and I do not
  have particular issues with it.  The main issue is that some key limitations
  are not discussed properly, see below.  The experiments are rather thorough
  and consider a number of research questions, competitors, ablations, ...

---

> ### Author Response · Authors · 2024-07-03
> **We appreciate your feedback and here is our responses!**
>
> We sincerely thank the Reviewer for the valuable and insightful feedback!
>
> Below, please see our responses to your comments and questions:

---

> ### Author Response · Authors · 2024-07-03
> **Regarding Weaknesses 1 in Presentation**
>
> > The message comprises two disconnected pieces: 1) showing explanations for the top-K classes aids human decision making, 2) reranking predictions using NNs helps machine performance. This is not a big deal, but a more streamlined message would have helped.
>
>
> Thank you for the suggestion!
> We want to note that a unified take-away from our work is that `showing PCNN explanations for the top-K classes improves both human and AI accuracy`.
>
> **PCNN explanations help improve AI accuracy**: PCNN explanations help train the fine-grained image comparator S (the AI), which is later used in the re-ranking algorithm CxS to improve the overall top-1 classification accuracy.
>
> Yet, after re-reading the paper, we agree with the Reviewer that the message could be more streamlined.
> To connect the two pieces better, we added a message in the beginning of Sec 4. Results (page 6) of the latest revision as:
>
> `
> In this section, we will show that PCNN explanations improve both AI and human accuracy. Regarding
> improved AI accuracy, we demonstrate that PCNN explanations can be used to both train fine-grained image
> comparator S and to re-rank then correct wrong predictions of a pretrained classifier C. For improved human
> accuracy, we show that when shown PCNN explanations, humans improve their accuracy in distinguishing
> between correct and incorrect predictions by almost +10 points.
> `

---

> ### Author Response · Authors · 2024-07-03
> **Regarding Weaknesses 2 in Presentation**
>
> > Another issue is the method name is not used consistently throughout the text. For instance, it is sometimes called C \times S, sometimes PoE; sometimes PCNN is an architecture, sometimes a new type of explanation; this is a bit confusing. I would prefer if the authors used PCNN everywhere, for simplicity.
>
> We agree with this point and to mitigate the confusion, we will use the term `CxS` consistently throughout the writing to refer to the architecture. To clarify, PCNN is a variant of nearest-neighbor explanations. We have also made sure to use the term `PCNN` consistently to refer to the explanations in our latest revision.
>
> We acknowledge that PCNN may be easily confused with the deep learning architecture CNN (convolutional neural networks), which also appears multiple times in our writing. If the Reviewer sees this as a potential issue for readability, we could add a footnote to clarify this.

---

> ### Author Response · Authors · 2024-07-03
> **Regarding Requested Changes 1 (1/2)**
>
> > The authors are upfront about the fact that PCNN requires training data at test time (like all kNN-based predictors), but do not list this as an actual limitation in Sec 5, while I think it is.
> There they mention run-time of PCNN is longer than other competitors, but the main issue is PCNN requires the training data to be available in entirety -- or, at least, the experiments do not study the impact of reducing the sice of the training data on inference-time performance. This is a clear downside compared to, say, ProtoPNets, which memorize relevant (part) prototypes instead.
> So there exists a clear trade-off between space and time requirements and prediction improvements (which is substantial but not huge to begin with, usually in the order of 2-3% top-1 accuracy over the runner-ups, at least according to Tables 3-5).
>
>
> Thank you for bringing up this excellent point! Obviously, requiring the whole training data at test time is a significant downside.
> The comment from the Reviewer encouraged us to investigate the impact of reducing the size of the training data on inference-time performance.
>
> Below, we present the experimental data on CUB-200 and Dogs-120:
>
> | Dataset   | % Data | Samples per Class | Top-1 Acc(%) | Runtime (s) |
> |-----------|--------|-------------------|--------------|-------------|
> | CUB-200   | 100%   | 30                | 88.43        | 64.55       |
> | CUB-200   | 50%    | 15                | 88.26        | 59.70       |
> | CUB-200   | 33%    | 10                | 88.19        | 58.08       |
> | Dogs-120  | 100%   | 100               | 86.27        | 87.18       |
> | Dogs-120  | 50%    | 50                | 86.32        | 71.02       |
> | Dogs-120  | 33%    | 33                | 86.42        | 65.52       |
>
> **_The top-1 accuracy and runtime of CxS on CUB-200 and Dogs-120 for different sizes of training data during inference. Runtime was computed over 1000 samples, similar to the setup in Appendix E._**
>
> **Findings**:
> - We found that reducing the size of the training data has little-to-no impact on the inference-time performance.
> - When keeping the same accuracy, we can reduce the runtime by `10%` on CUB-200 and `24.9%` on Dogs-120 by reducing the training set to `33%` of the original size.
>
> We added this experiment in Sec. E.2 of the latest version.

---

> ### Author Response · Authors · 2024-07-03
> **Regarding Requested Changes 1 (2/2)**
>
> To further narrow the gap in runtime between our method vs. other interpretable classifiers, we also attempted to reduce the overhead introduced by the re-ranking process by reducing the number of queries to the image comparator S.
> Currently, we are always examining the top-`K` (with `K=10`) most probable classes.
> Yet, there always exists classes assigned a `< 1%` probability by ResNet50 (`Fig. 5` in PCNN submission) and are not in the top-1 after re-ranking.
> Reducing the number of `K` can save a lot of computation at a minimal cost of accuracy.
>
> To verify this, we run an experiment for CUB-200 where we instead of re-ranking the whole **top-10**, we only re-rank the classes that have a probability `>= 1%` assigned by the base classifier C.
>
> **Findings**:
> - The PoE model performance on CUB-200 drops very marginally by `only 0.08%` (from `88.59%` → `88.51%`).
> - However, the number of queries to the image comparator S was reduced by approx. `4x` (from `10` to just about `2.5` queries/image).
> This leads to a `2.5x` speedup in the overall runtime of the PoE system (from `64.55` seconds to `28.95` seconds per 1000 images), as shown in the following Table.
>
> | Model                        |   Time (s)   | Top-1 Acc (%) |
> |:----------------------------:|:------------:|:-------------:|
> | RN50 xS (before)                      | 64.55 ± 0.35 |     88.59     |
> | RN50 xS (after) | 28.95 ± 0.11 |     88.51     |
>
> **_The run-time of PoE on 1,000 queries on one Nvidia V100 GPU._**
>
> We added this experiment in Sec. E.1 of the latest version!

---

> ### Author Response · Authors · 2024-07-03
> **Regarding Requested Changes 2**
>
> > I don't think the results of the user study are reported appropriately in the introduction.
> Bottom line: showing more NNs to users makes them more skeptical, meaning they end up underestimating machine perforance.
> This should be clearly stated in the introduction, at the bare minimum.
> Instead, the authors focus on the benefits of PCNN only, and write: "A 60-user study finds that, compared to showing top-1 class examples, PCNN improves user performance on the distinction task by almost +10 points (pp) (54.55% vs. 64.58%) on CUB-200 (Sec. 4.6)." I don't think this is entirely fair and the text should be amended. This should also be listed in the Limitations section.
>
> Thank you for this thoughtful comment!
>
> We agree with the Reviewer that the introduction should clearly state the bottom line of the user study.
> After careful revision, we attribute the improvement in user accuracy to the reduced over-reliance (an established term) [1] where users tend to rely less on machine predictions, particularly when the model is wrong, when shown PCNN explanations.
> This observation also aligns with the findings of Buçinca et al. [2] and Bansal et al. [3] that reduced over-reliance helps improve AI-assisted decision-making.
>
> We added this bottom-line message to the latest version in 3 venues:
>
> In Abstract:
> `
> Also, a human study finds that showing lay users our probable-class nearest neighbors (PCNN)
> reduces over-reliance on AI, thus improving their decision accuracy over prior work which
> only shows only the top-1 class examples.
> `
>
> In Introduction:
> `
> A 60-user study finds that PCNN explanations, compared with top-1 class examples, reduce over-
> reliance on AI, thus improving user performance on the distinction task by almost 10 points (54.55%
> vs. 64.58%) on CUB-200 (Sec. 4.6).
> `
>
> In Sec 4.6. (page 12):
> `
> Our finding aligns with the literature that showing explanations helps users reduce
> over-reliance on machine predictions Buçinca et al. (2021); Schemmer et al. (2023); Chen et al. (2023a).
> `
>
> 1. Appropriate reliance on AI advice: Conceptualization and the effect of explanations, IUI (2023).
> 2. To trust or to think: cognitive forcing functions can reduce overreliance on AI in AI-assisted decision-making, CSCW (2021).
> 3. Does the whole exceed its parts? the effect of ai explanations on complementary team performance, CHI (2021).

---

> ### Author Response · Authors · 2024-07-03
> **Regarding Requested Changes 3**
>
> > The construction of the training set for S assumes C is already reasonably high-quality: is this always a reasonable assumption? Please add this to the Limitations section too.
>
>
> Our answer is: Classifiers C is NOT strictly required to be `reasonably high-quality`.
>
> To support that, we tested three cases: **(1)** using high-performing classifiers C (e.g. >= 80% top-1 acc), **(2)** using low-performing classifiers C (e.g. ~ 60% top-1 acc), and **(3)** excluding classifiers C from sampling process.
>
> **(1)** We perform sampling using `high-performing classifiers` C (e.g. ResNet-50 scores `85.83%` on CUB-200, `89.73%` on Cars-196, and `85.82%` on Dogs-120) to train S and report numbers in Table 1 of the latest revision.
> We often see improvements from `1-3 points` when comparing CxS to C.
>
> **(2)** We also test the case where we use `low-performing` classifiers C (e.g. ResNet classifiers pretrained on ImageNet score `60-63%` on CUB-200) and report numbers in Table 1.
> It is interesting that we see even much bigger improvements in this case, e.g. up to almost `12 points` on CUB-200.
>
> **(3)** We also demonstrated in the latest version (Sec 4.2) that we can *exclude classifier C in sampling*. Please refer to the paragraph **Hard negatives are more useful than easy, random negatives in training S**.
> In this experiment, the negative samples are randomly chosen from non-ground-truth classes, and the positives are the ground-truth class, known from training set annotations.
> This sampling still yield positive improvement in the top-1 accuracy for a CUB-200 iNaturalist-pretrained ResNet-50 from `85.85% → 86.55%` (+0.70%).

---

> ### Author Response · Authors · 2024-07-03
> **Regarding Requested Changes 4**
>
> > An analysis of errors introduced by the reranking step would have been useful.
>
>
> Thank you for this suggestion!
>
> To analyze where reranking failed, we visualized `700 CUB-200` samples that were `misclassified` by our re-ranking method (CxS 88.59% in Table. 1). You can also access them [here](https://drive.google.com/drive/folders/1CJRAmuoBSLQQ2ES0dmCbiTDTCWZvewzM?usp=sharing).
>
> After manually inspecting the samples, we found that the majority of the misclassifications were due to the following reasons:
> 1. **Inter-class similarities**: The species from different classes look very similar to each other.
> 2. **Intra-class variations**: The species from the same class have large variations in appearance because of lighting or angles. This is expected because the birds are captured in the wild.
> 3. **Low initial confidence scores**: The initial classifier C assigns too low confidence scores to the correct class because of using softmax. Therefore, even if the comparator S assigns high confidence scores, the CxS score is not sufficient to be recognized as the top-1.
>
> We believe that both (1) and (2) are inherent to the dataset and not specific to our method. These observations are also supported by an error analysis in Sec I.2 of our submission. A possible remedy for this to explore in future work is data augmentation. We do provide preliminary evidence that data augmentation can benefit image comparator S in Sec. 4.2.
>
> Regarding (3), one solution could be re-normalizing the confidence scores of C (e.g. skip the softmax layer to make confidence scores less extreme). We leave this for future work.

---

> ### Author Response · Authors · 2024-07-03
> **Regarding Requested Changes 5**
>
> > p 4: "We empirically test K = {1, 3, 5, 10} and find K = 10 to be optimal." The fact that the optimal value is at the very end of the spectrum begs the question whether increasing K could improve performance further. Did the authors evaluate what happens for larger values of K? Clearly, increasing K would not be ideal for human decision making, but it should not consistute for PCNNs proper.
>
>
> Thank you for this valuable comment!
> We re-run our method with different values of `K` and present the results in the below Table.
>
> | K  | CUB Perf (%)  | CUB Runtime (s) | Dogs Perf (%)         | Dogs Runtime(s) |
> |----|---------------|-----------------|-----------------------|-----------------|
> | 1  | 85.83         | 8.81            | 85.82                 | 8.81            |
> | 2  | 87.95 (+2.12) | 27.72           | 86.06 (+0.24)         | 50.35           |
> | 3  | 88.28 (+2.45) | 32.32           | 86.03 (+0.21)         | 54.96           |
> | 5  | 88.28 (+2.45) | 41.53           | 85.91 (+0.09)         | 64.16           |
> | 10 | 88.42 (+2.59) | 64.55           | 86.27 (+0.45)         | 87.18           |
> | 15 | 88.00 (+2.17) | 87.57           | 85.86 (+0.04)         | 110.20          |
>
> **_The top-1 accuracy and runtime of CxS on CUB-200 and Dogs-120 for different K values. Runtime was computed over 1000 samples, similar to the setup in Appendix E._**
>
> We tested with K values are `{1,2,3,5,10,15}` on CUB-200 and Dogs-120.
> We found that increasing `K` *does not only hurt the classification accuracy but also increases the runtime*.
> Still, using `K = 10` strikes the optimal balance between accuracy and runtime.

---

> ### Author Response · Authors · 2024-07-03
> **Regarding Requested Changes 6**
>
> > p 2 onwards: The authors say their model is a Product of Experts (PoE), based on the definition of Hinton, 1999. Reading through this reference, however, I get the impression that in PoEs the various distributions are conditionally independent given the input (for instance, in p 4 of Hinton '99, they state "the hidden states of different experts are conditionally independent given the data"). The same (conditional) independence assumption seems to be instrumental in more recent research on PoEs, see:
> Gordon et al., "Identifiability of Product of Experts Models." 2024.
> To me, conditional independence seems necessary to reinterpret the product of distributions as a factorization of a more complex joint distribution, which lies at the heart of PoEs.
> However, independence does not appear to hold for the CSP model. I would appreciate if the authors could clarify this point, and -- if independence is indeed a prerequisite of PoEs -- changed their wording accordingly.
>
>
> Thank you for this insightful comment!
>
> First, let us reiterate the definition of PoE:
> `A PoE model combines multiple probability distributions (experts) to form a more complex joint distribution, where each expert captures different aspects of the data, and their product forms the overall model.`
> Indeed, we agree with the Reviewer that conditional independence lies at the heart of PoEs because this enables factorization of distributions.
>
> Let's re-visit our `CxS` model:
>
> Model `C`: A pre-trained classifier producing a probability distribution over classes for a given input image.
>
> Model `S`: Image comparator, which compares an input image with its nearest neighbors and generates confidence scores.
>
> To verify if C and S can be considered experts in a PoE framework, we need to determine if their outputs are conditionally independent given the input data.
> Conditional independence implies that the probability distribution over classes from C should not influence the probability distribution from S, once we know the input image.
>
> However, the P(C) and P(S) are not conditionally independent given an input image.
> This is because C produces a class distribution, and S refines this distribution based on nearest neighbors. The confidence score S assigns can be influenced by the initial ranking from C.
> Due to the dependency between C and S, our model does not strictly follow the PoE framework's requirement.
>
> Given that, we changed the wording properly throughout in the latest version. We truly appreciate the Reviewer for pointing out this discrepancy!
>
> We also added the connection between our model to PoE into Sec. 2 (Related Work) at page 3 that:
>
> `
> A key difference from standard PoE and boosting techniques is that we leverage training-set examples (PCNN) at the test time of S, improving CxS model accuracy further over the baseline C.
> Also, our model does not strictly follow the PoE framework's requirement of conditional independence between experts because the confidence scores that image compartor S assigns to most-probable classes can be influenced by the initial ranking from C.
> However, our CxS framework is similar to the PoE (Eq.1 in [a]) in the sense that both C and S has the veto power and the final probability is the product of both probabilities.
> `
>
> [a] Generalized Product of Experts for Automatic and Principled Fusion of Gaussian Process Predictions

---

> ### Author Response · Authors · 2024-07-03
> **Regarding Broader Impact**
>
> > Broader impact should briefly discuss the potential impact of manipulating (in the case of PCNN, lowering) user confidence in machine predictions in, especially in time critical high-stakes scenarios.
>
> Thank you for the suggestion!
>
> Indeed, trust calibration is a critical aspect of human-AI interaction, especially in high-stakes scenarios per the Reviewer's comment.
> We included a discussion on the potential impact of manipulating user confidence in machine predictions in the Sec. 6. Discussion and Conclusion in the latest revision.
>
> `
> An important aspect to consider is that while PCNN provides a comprehensive explanation by presenting
> multiple top-class predictions, it is crucial to ensure this does not reduce user confidence to a degree that
> negatively affects decision-making, especially in time-critical high-stakes scenarios. Balancing detailed
> information with user confidence is vital for effective human-machine collaboration and reliable decision-
> making
> `

---

> ### Author Response · Authors · 2024-07-23
> **Thank you for the acknowledgement!**
>
> We sincerely thank Reviewer `bLsM` for the thoughtful feedback.
>
> We will include the discussion here properly into the final version!

---

### Review · Reviewer_96cH · 2024-06-20

**Summary Of Contributions:**

The study focuses on leveraging nearest neighbors (NN) to enhance predictions made by a pretrained classifier. The authors propose a method that employs an image comparator, labeled as S, which serves two crucial functions. Firstly, it compares the input image with NN images from the top-K most probable classes. Secondly, it utilizes the output scores from S to weigh the confidence scores of the pretrained classifier, denoted as C. Through this approach, the researchers consistently demonstrate improved accuracy in fine-grained image classification across datasets such as CUB-200, Cars-196, and Dogs-120. Moreover, the paper incorporates a human study wherein lay users are presented with the nearest neighbors of probable classes instead of solely the top-1 class examples.

**Audience:**

Yes

**Claims And Evidence:**

Yes

**Requested Changes:**

1.  Include a discussion on potential strategies to address the challenge of extended inference time.

2. Can you determine an optimal value for K by considering the unique characteristics of each query or input, in order to enhance the balance between accuracy and runtime efficiency?

3. Is it possible to implement incremental learning techniques to update the comparator S over time with new data?

**Strengths And Weaknesses:**

Strengths:

1.  The proposed method, PCNN, enhances the accuracy of a fine-grained image classification system without the need to retrain the underlying black-box classifier.
2. The PCNN method achieves state-of-the-art classification accuracy on benchmark datasets such as CUB-200, Cars-196, and Dogs-120.
3. The human study conducted in the research reveals that presenting PCNN to users improves their decision-making accuracy compared to the conventional practice of displaying only the top-1 class examples.

Weaknesses:

1. When compared to other existing classifiers, the propsed method is slower than k-NN and prototypical part-based classifiers but quicker than CHM-Corr and EMD-Corr re-rankers. Although it is in line with existing methods, its slower speed could be a drawback in situations where efficiency is crucial.

---

> ### Author Response · Authors · 2024-07-03
> **Thank you and here is our rebuttal!**
>
> We sincerely thank the Reviewer for the feedback and constructive suggestions!
> Please find our responses to each of your concerns below.

---

> ### Author Response · Authors · 2024-07-03
> **Regarding Weaknesses 1 and Requested Changes 1 (1/2)**
>
> > (Weaknesses 1) When compared to other existing classifiers, the propsed method is slower than k-NN and prototypical part-based classifiers but quicker than CHM-Corr and EMD-Corr re-rankers. Although it is in line with existing methods, its slower speed could be a drawback in situations where efficiency is crucial.
>
> > (Requested Changes 1) Include a discussion on potential strategies to address the challenge of extended inference time.
>
> We would like to respond to both of these concerns together because they are closely related.
>
> The total runtime of our CxS is computed by:
>
> `T = T_RN50 + T_kNN + T_S` (Equation. 3 in our paper)
>
> where `T_RN50` is the time to do inference with pretrained classifier C (here RN50), `T_kNN` is the time to retrieve nearest neighbors, and `T_S` is the time to compute the similarity scores using image comparator S.
> Here, we show that we can significantly reduce `T_kNN` and `T_S` while maintaining the accuracy of our method:
>
> **(1)** By reducing `T_kNN` by shrinking training set during inference
>
> Requiring the whole training data at test time is a significant downside and it slows down the process of retrieving nearest neighbors.
>
> Therefore, we attempt to reduce the training set from `100% → 50% → 33%` of the original size and observe the effect on the accuracy and runtime.
> Here is the experimental data on CUB-200 and Dogs-120:
>
> | Dataset   | % Data | Samples per Class | Top-1 Acc(%) | Runtime (s) |
> |-----------|--------|-------------------|--------------|-------------|
> | CUB-200   | 100%   | 30                | 88.43        | 64.55       |
> | CUB-200   | 50%    | 15                | 88.26        | 59.70       |
> | CUB-200   | 33%    | 10                | 88.19        | 58.08       |
> | Dogs-120  | 100%   | 100               | 86.27        | 87.18       |
> | Dogs-120  | 50%    | 50                | 86.32        | 71.02       |
> | Dogs-120  | 33%    | 33                | 86.42        | 65.52       |
>
> **_The top-1 accuracy and runtime of CxS on CUB-200 and Dogs-120 for different sizes of training data during inference. Runtime was computed over 1000 samples, similar to the setup in Appendix E._**
>
>
> **Findings**:
> - We found that reducing the size of the training data has little-to-no impact on the inference-time performance.
> - When keeping the same accuracy, we can reduce the runtime by `10%` on CUB-200 and `24.9%` on Dogs-120 by reducing the training set to `33%` of the original size.
>
> We added this experiment in Sec. E.2 of the latest version.

---

> ### Author Response · Authors · 2024-07-03
> **Regarding Weaknesses 1 and Requested Changes 1 (2/2)**
>
> **(2)** By Optimizing `T_S` with reduced number of image comparisons done by S
>
> Currently, we are setting this to `K = 10` as we always examine the **top-10** most probable classes.
> Yet, there always exists classes receiving `< 1%` by ResNet50 (see Fig. 5 in PCNN submission), which are likely never to be the top-1 after re-ranking.
> Reducing the number of `K` can save a lot of computation at a minimal cost of accuracy.
>
> We run an experiment for CUB-200 where we instead of re-ranking the whole **top-10**, we only re-rank the classes that have a probability `>= 1%` assigned by the base classifier C.
>
> **Our findings**:
> - The `CxS` model accuracy on CUB-200 drops very marginally by `only 0.08%` (from `88.59%` → `88.51%`).
> - However, the number of queries to the image comparator S was reduced by approx. `4x` (from `10` to just about `2.5` queries/image).
> This leads to a `2.5x` speedup in the overall runtime of the CxS system (from `64.55` seconds to `28.95` seconds per 1000 images), as shown in the following Table.
>
> |               Model                |   Time (s)   | Top-1 Acc (%) |
> |:----------------------------------:|:------------:|:-------------:|
> |          RN50 xS (before)          | 64.55 ± 0.35 |     88.59     |
> |          RN50 xS (after)           | 28.95 ± 0.11 |     88.51     |
>
> **_The run-time of CxS on 1,000 queries on one Nvidia V100 GPU._**
>
> We added this experiment in Sec. E.1 of the latest version.
>
> We also include a discussion on potential strategies to further speed up runtime in response to the Reviewer [`g7wm`](https://openreview.net/forum?id=OcFjqiJ98b&noteId=XuB3bY6d9q).
> In this response, we suggest to leverage GPU to speed up the nearest neighbor retrieval algorithm and use built-in options from `faiss` library to speed up the similarity computation.

---

> ### Author Response · Authors · 2024-07-03
> **Regarding Requested Changes 2**
>
> > (Requested Changes 2) Can you determine an optimal value for K by considering the unique characteristics of each query or input, in order to enhance the balance between accuracy and runtime efficiency?
>
>
> Thank you for your suggestion!
>
> Predicting an optimal K for each input image is an interesting idea yet requires building a separate predictive model that can be useful for many tasks.
> We are not aware of an existing tool that can already do this and therefore leave this idea for future work.
>
> Yet, we agree with your concern of long runtime (due to a large K) and are proposing to reduce the K from `10` down to around `2.5` per query by using only the top labels with confidence scores > `0.01` assigned by classifier C [here](https://openreview.net/forum?id=OcFjqiJ98b&noteId=ZDKb4mPz0V).
> We found that this can significantly reduce the runtime of `CxS` from `64.55` secs to `28.95` secs (around `2.5x`) with a marginal drop (`0.08%`) in accuracy.

---

> ### Author Response · Authors · 2024-07-03
> **Regarding Requested Changes 3**
>
> > Is it possible to implement incremental learning techniques to update the comparator S over time with new data?
>
> Yes! It is possible to update S as our framework takes in two separate models: C and S.
>
> We don’t believe that our framework imposes any specific constraints against continual learning.
> With popular incremental learning techniques, such as replay [1], pseudo-rehearsal[2] , or regularization [3], future work can explore approaches to improve S over time by adding more data.
>
> 1. Continual learning with deep generative replay, NeurIPS 2017
> 2. ICARL: Incremental classifier and representation learning, CVPR 2017
> 3. Learning without Forgetting, ECCV 2016

---

### Decision · Action_Editor_53jp · 2024-07-27

**Recommendation:** Accept as is

**Comment:**

The manuscript proposes a method based on Nearest Neighbor Explanations for top-K predicted classes to improve human decision making accuracy as well as the model's accuracy. Experiments as well as a human study demonstrate that the method is effectie on several fine-grained image classification datasets.

**Audience:**

Yes

**Claims And Evidence:**

Yes